# Learning Single Index Models with Diffusion Priors

**Anqi Tang** [* 1]  **Youming Chen** [* 1]  **Shuchen Xue** [2 3]  **Zhaoqiang Liu** [1]

## Abstract

Diffusion models (DMs) have demonstrated remarkable ability to generate diverse and high-quality images by efficiently modeling complex data distributions. They have also been explored as powerful generative priors for signal recovery, resulting in a substantial improvement in the quality of reconstructed signals. However, existing research on signal recovery with diffusion models either focuses on specific reconstruction problems or is unable to handle nonlinear measurement models with discontinuous or unknown link functions. In this work, we focus on using DMs to achieve accurate recovery from semi-parametric single index models, which encompass a variety of popular nonlinear models that may have *discontinuous* and *unknown* link functions. We propose an efficient reconstruction method that only requires one round of unconditional sampling and (partial) inversion of DMs. Theoretical analysis on the effectiveness of the proposed methods has been established under appropriate conditions. We perform numerical experiments on image datasets for different nonlinear measurement models. We observe that compared to competing methods, our approach can yield more accurate reconstructions while utilizing significantly fewer neural function evaluations.

## 1. Introduction

The objective of compressed sensing is to accurately recover the underlying high-dimensional sparse signal from a small number of linear measurements, with the measurement model as follows (Candes & Wakin, 2008; Foucart &

---
[*]Equal contribution [1]University of Electronic Science and Technology of China [2]University of Chinese Academy of Sciences [3]Academy of Mathematics and Systems Science, CAS. Correspondence to: Zhaoqiang Liu <zqliu12@gmail.com>.

*Proceedings of the $42^{nd}$ International Conference on Machine Learning*, Vancouver, Canada. PMLR 267, 2025. Copyright 2025 by the author(s).

Rauhut, 2013; Scarlett & Cevher, 2021):

$$\boldsymbol{y} = \mathbf{A}\boldsymbol{x}^* + \boldsymbol{e}, \tag{1}$$

where $\mathbf{A} \in \mathbb{R}^{m \times n}$ is the measurement matrix, $\boldsymbol{x}^* \in \mathbb{R}^n$ is the signal to be recovered, $\boldsymbol{e} \in \mathbb{R}^m$ is the additive noise vector, and $\boldsymbol{y} \in \mathbb{R}^m$ is the observed vector.

Although the linear measurement model utilized in compressed sensing can serve as a good testbed for demonstrating conceptual phenomena, in many real-world problems, it may not be justifiable or even feasible. For instance, the binary measurement model employed in 1-bit compressed sensing (Boufounos & Baraniuk, 2008) has attracted substantial interest. This is due to its cost-effective and efficient hardware. It has also been shown that 1-bit compressed sensing is robust against certain nonlinear distortions (Laska & Baraniuk, 2012). The limitations of the linear data model have prompted the study of general nonlinear measurement models. Among these, the semi-parametric single index model (SIM) is arguably the most popular (Horowitz, 2009). The SIM models the observations as follows:

$$\boldsymbol{y} = f(\mathbf{A}\boldsymbol{x}^*), \tag{2}$$

where $\mathbf{A}$ consists of i.i.d. standard Gaussian entries, and $f : \mathbb{R} \to \mathbb{R}$ is an *unknown* and possibly random nonlinear link function that is applied element-wise. The aim is to estimate the signal $\boldsymbol{x}^*$ using the knowledge of $\mathbf{A}$ and $\boldsymbol{y}$, despite the *unknown* nonlinear link $f$. It is well-known that in the SIM, $\boldsymbol{x}^*$ is generally not identifiable (Plan & Vershynin, 2016), since any scaling of $\boldsymbol{x}^*$ can be incorporated into the unknown $f$. Consequently, it is common to impose the identifiability constraint $\|\boldsymbol{x}^*\|_2 = 1$.

To enable efficient and reliable recovery of the underlying signal $\boldsymbol{x}^*$, the predominant approach is to introduce structural modeling assumptions, such as sparse and conventional generative priors. For instance, the research works following (Bora et al., 2017) typically achieve signal reconstruction under the assumption that the underlying signal lies within the range of a variational autoencoder (VAE) or a generative adversarial network (GAN) (Van Veen et al., 2018; Hand et al., 2018; Heckel & Hand, 2019; Wu et al., 2019; Asim et al., 2020; Ongie et al., 2020; Whang et al., 2020; Jalal et al., 2021b; Nguyen et al., 2021; Liu et al., 2021b; 2022a; Genzel et al., 2022; Liu et al., 2024).

Compared to conventional generative models like VAEs and GANs, diffusion models (DMs) (Sohl-Dickstein et al., 2015; Song & Ermon, 2019; Ho et al., 2020) have recently demonstrated their superiority in terms of robust generative capabilities and learning complex data distributions (Dhariwal & Nichol, 2021). DMs have been integrated into a wide variety of domains, leading to transformative progress and reshaping traditional approaches in areas such as vision and audio generation (Saharia et al., 2022b; Kong et al., 2021), and reinforcement learning (Chi et al., 2023; Hansen-Estruch et al., 2023). In particular, when it comes to addressing signal recovery problems, diffusion models offer a more flexible and robust solution, facilitating high-quality reconstructions and improved performance (Kawar et al., 2022; Whang et al., 2022; Saharia et al., 2022c;a; Alkan et al., 2023; Chung et al., 2023a; Cardoso et al., 2023; Wang et al., 2023; Chung et al., 2023b; Feng & Bouman, 2023; Rout et al., 2023; Wu et al., 2023; Aali et al., 2024; Chung et al., 2024; Dou & Song, 2024; Song et al., 2024; Wu et al., 2024; Sun et al., 2024).

Inspired by the strong capabilities of DMs in diverse real-world applications, in this study, we investigate the application of DMs to learn SIMs.

## 1.1. Related Work

Relevant works on signal recovery with conventional generative models (such as GANs and VAEs) have shown that learned priors can drastically reduce sample complexity in compressed sensing and related problems. A comprehensive discussion of diffusion-based and conventional generative approaches is provided in Appendix A. In the following, we highlight diffusion-based signal recovery methods.

**Signal recovery with diffusion models:** Diffusion models (DMs) have advanced signal recovery by modeling complex data distributions, with two main paradigms: task-specific and pre-trained approaches. Task-specific DMs are trained for particular problems, such as super-resolution (Saharia et al., 2022c) or deblurring (Ren et al., 2023). Pre-trained DMs, used without task-specific training, guide reconstruction via score estimation (e.g., (Feng et al., 2023)) or adapt frameworks like DDPM for linear measurements (e.g., DDRM (Kawar et al., 2022)). MCG (Chung et al., 2022a) and ΠGDM (Song et al., 2023) are mainly designed for the linear setting. Additionally, ΠGDM can be extended to certain nonlinear settings (where the link function is known) by leveraging a combination of pseudoinverse operations and nonlinear transformations. DPS (Chung et al., 2023b) and DAPS (Zhang et al., 2024a) are applicable in nonlinear settings as well, also primarily under the assumption that the link function is known.

However, the works mentioned above either concentrate on specific linear signal recovery problems or are incapable of handling nonlinear measurement models with *discontinuous* or *unknown* link functions. Perhaps most relevant to the present paper, the work (Meng & Kabashima, 2022) uses SGMs to learn the underlying distribution of natural signals. By estimating the conditional score function, it enables better reconstruction quality when dealing with quantized compressed sensing problems. Nevertheless, the approach in (Meng & Kabashima, 2022) is restricted to quantized measurements and is relatively slow in reconstruction. Even if utilizing the knowledge of the link function, it requires thousands of neural function evaluations (NFEs) to achieve a reasonably accurate recovery.

## 1.2. Contributions

The main contributions of this paper are threefold:

- We propose an efficient method for accurately recovering the underlying signal from nonlinear observations of SIMs by utilizing diffusion priors. Our approach *does not require the knowledge of the link function*, and it only requires one round of unconditional sampling and (partial) inversion of DMs, resulting in a relatively small number of NFEs for the entire process.

- Based on heuristic theoretical results, we opt to initiate the inversion of DMs from an intermediate time rather than performing the full inversion of DMs. Under suitable conditions, we provide a theoretical analysis on the effectiveness of the corresponding approach. Additionally, we empirically observe that carrying out partial inversion results in significantly better reconstructions compared to performing full inversion.

- To validate the effectiveness of our method, we conduct numerical experiments on image datasets under distinct nonlinear measurement models. Notably, in the noisy 1-bit measurement setting, our approach not only achieves substantially faster computational performance but also outperforms competing methods in reconstruction accuracy, even when these methods explicitly incorporate knowledge of the link function.

## 1.3. Notation

We use upper and lower case boldface letters to denote matrices and vectors respectively. For any $M \in \mathbb{N}$, we use the shorthand notation $[M] = \{1, 2, \ldots, M\}$, and we use $\boldsymbol{I}_M$ to denote the identity matrix in $\mathbb{R}^{M \times M}$. Given two sequences of real values $\{a_i\}$ and $\{b_i\}$, we write $a_i = O(b_i)$ if there exists an absolute constant $C_1$ and a positive integer $i_1$ such that for any $i > i_1$, $|a_i| \leq C_1 b_i$. We use $\mathcal{S}^{n-1}$ to denote the unit sphere in $\mathbb{R}^n$, i.e., $\mathcal{S}^{n-1} = \{\boldsymbol{s} \in \mathbb{R}^n : \|\boldsymbol{s}\|_2 = 1\}$. For a generator $G : \mathbb{R}^n \to \mathbb{R}^n$, we use $\mathcal{R}(G)$ to denote the range of $G$, i.e., $\mathcal{R}(G) = \{G(\boldsymbol{z}) : \boldsymbol{z} \in \mathbb{R}^n\}$.

## 2. Preliminaries for Diffusion Models

Diffusion Models (DMs) define a forward process $\{\boldsymbol{x}_t\}_{t \in [0,T]}$ starting from $t = 0$ and $\boldsymbol{x}_0$. For any $t \in [0, T]$, the distribution of $\boldsymbol{x}_t$ conditioned on $\boldsymbol{x}_0$ satisfies the following equation:

$$q_{0t}(\boldsymbol{x}_t|\boldsymbol{x}_0) = \mathcal{N}(\alpha_t \boldsymbol{x}_0, \sigma_t^2 \boldsymbol{I}_n). \tag{3}$$

Here, $\alpha_t$ and $\sigma_t$ are non-negative, differentiable functions possessing bounded derivatives. Moreover, $\alpha_t$ is monotonically non-increasing with $\alpha_0 = 1$, while $\sigma_t$ is monotonically increasing. It has been demonstrated that the following stochastic differential equation (SDE) has the same transition distribution $q_{0t}(\boldsymbol{x}_t|\boldsymbol{x}_0)$ for any $t \in [0, T]$ (Song et al., 2021b; Kingma et al., 2021):

$$\mathrm{d}\boldsymbol{x}_t = f(t)\boldsymbol{x}_t \mathrm{d}t + g(t)\mathrm{d}\boldsymbol{w}_t, \quad \boldsymbol{x}_0 \sim q_0(\boldsymbol{x}_0), \tag{4}$$

where $\boldsymbol{w}_t \in \mathbb{R}^n$ is the standard Wiener process, and $q_t$ represents the marginal distribution of $\boldsymbol{x}_t$. Moreover, $f(t)$ is a scalar function known as the drift coefficient and $g(t)$ is a scalar function called the diffusion coefficient, which are specified as follows:

$$f(t) = \frac{\mathrm{d}\log\alpha_t}{\mathrm{d}t}, \quad g^2(t) = \frac{\mathrm{d}\sigma_t^2}{\mathrm{d}t} - 2\frac{\mathrm{d}\log\alpha_t}{\mathrm{d}t}\sigma_t^2. \tag{5}$$

Under certain regularity conditions, Song et al. (2021b) demonstrate that the forward process in Eq. (4) has an equivalent reverse process running from time $T$ to 0, which starts with the marginal distribution $q_T(\boldsymbol{x}_T)$:

$$\mathrm{d}\boldsymbol{x}_t = [f(t)\boldsymbol{x}_t - g^2(t)\nabla_{\boldsymbol{x}}\log q_t(\boldsymbol{x}_t)]\mathrm{d}t + g(t)\mathrm{d}\bar{\boldsymbol{w}}_t, \tag{6}$$

where $\bar{\boldsymbol{w}}_t \in \mathbb{R}^n$ represents the Wiener process in reverse time, and $\nabla_{\boldsymbol{x}}\log q_t(\boldsymbol{x}_t)$ stands for the time-varying score function.

For more efficient sampling, Song et al. (2021b) further show that the following probability flow ordinary differential equation (ODE) possesses the same marginal distribution at each time $t$ as that of reverse SDE in Eq. (6):

$$\frac{\mathrm{d}\boldsymbol{x}_t}{\mathrm{d}t} = f(t)\boldsymbol{x}_t - \frac{1}{2}g^2(t)\nabla_{\boldsymbol{x}}\log q_t(\boldsymbol{x}_t), \quad \boldsymbol{x}_T \sim q_T(\boldsymbol{x}_T). \tag{7}$$

The only unknown term in Eq. (7) is the time-varying score function $\nabla_{\boldsymbol{x}}\log q_t(\boldsymbol{x}_t)$. In practice, DMs typically estimate the scaled score function $-\sigma_t \nabla_{\boldsymbol{x}}\log q_t(\boldsymbol{x}_t)$ by using a noise prediction network $\boldsymbol{\epsilon}_{\boldsymbol{\theta}}(\boldsymbol{x}_t, t)$, and optimizes the parameter $\boldsymbol{\theta}$ through minimizing the following objective:

$$\int_0^T \mathbb{E}_{\boldsymbol{x}_0 \sim q_0}\mathbb{E}_{\boldsymbol{\epsilon} \sim \mathcal{N}(\boldsymbol{0}, \boldsymbol{I}_n)}\left[\|\boldsymbol{\epsilon}_{\boldsymbol{\theta}}(\alpha_t \boldsymbol{x}_0 + \sigma_t \boldsymbol{\epsilon}, t) - \boldsymbol{\epsilon}\|_2^2\right]\mathrm{d}t. \tag{8}$$

By substituting $\nabla_{\boldsymbol{x}}\log p_t(\boldsymbol{x}_t) = -\boldsymbol{\epsilon}_{\boldsymbol{\theta}}(\boldsymbol{x}_t, t)/\sigma_t$ into Eq. (7) and taking advantage of its semi-linear structure, we obtain the following numerical integration formula (Lu et al., 2022a; Zhang & Chen, 2022):

$$\boldsymbol{x}_t = e^{\int_s^t f(\tau)\mathrm{d}\tau}\boldsymbol{x}_s + \int_s^t \left(e^{\int_\tau^t f(r)\mathrm{d}r} \cdot \frac{g^2(\tau)}{2\sigma_\tau} \cdot \boldsymbol{\epsilon}_{\boldsymbol{\theta}}(\boldsymbol{x}_\tau, \tau)\right)\mathrm{d}\tau. \tag{9}$$

Alternatively, we can use a data prediction network $\boldsymbol{x}_{\boldsymbol{\theta}}(\boldsymbol{x}_t, t)$ that satisfies the condition $\boldsymbol{\epsilon}_{\boldsymbol{\theta}}(\boldsymbol{x}_t, t) = (\boldsymbol{x}_t - \alpha_t \boldsymbol{x}_{\boldsymbol{\theta}}(\boldsymbol{x}_t, t))/\sigma_t$ (Kingma et al., 2021; Salimans & Ho, 2022). By substituting Eq. (5) into Eq. (9) and using the transition between $\boldsymbol{\epsilon}_{\boldsymbol{\theta}}$ and $\boldsymbol{x}_{\boldsymbol{\theta}}$, we obtain the following integration formula (Lu et al., 2022b):

$$\boldsymbol{x}_t = \frac{\sigma_t}{\sigma_s}\boldsymbol{x}_s + \sigma_t \int_{\lambda_s}^{\lambda_t} e^\lambda \hat{\boldsymbol{x}}_{\boldsymbol{\theta}}(\hat{\boldsymbol{x}}_\lambda, \lambda)\mathrm{d}\lambda, \tag{10}$$

where $\lambda_t := \log(\alpha_t/\sigma_t)$ is strictly decreasing with respect to $t$ and has an inverse function $t_\lambda(\cdot)$. Thus, $\boldsymbol{x}_{\boldsymbol{\theta}}(\boldsymbol{x}_t, t)$ can be written as $\boldsymbol{x}_{\boldsymbol{\theta}}(\boldsymbol{x}_{t_\lambda(\lambda)}, t_\lambda(\lambda)) = \hat{\boldsymbol{x}}_{\boldsymbol{\theta}}(\hat{\boldsymbol{x}}_\lambda, \lambda)$.

### 2.1. The Sampling Process of Diffusion Models

The sampling process in DMs gradually denoises a noise vector to an image vector that approximately follows the same distribution as the training data. More precisely, if we divide the time interval $[\epsilon, T]$[1] into $N$ sub-intervals with $\epsilon = t_N < t_{N-1} < \ldots < t_1 < t_0 = T$, and use a first-order numerical scheme for Eq. (10), for $i \in [N]$, we can derive the following iterative formula for the transition from time $t_{i-1}$ to $t_i$:

$$\tilde{\boldsymbol{x}}_{t_i} = \frac{\sigma_{t_i}}{\sigma_{t_{i-1}}}\tilde{\boldsymbol{x}}_{t_{i-1}} + \sigma_{t_i}\left(\frac{\alpha_{t_i}}{\sigma_{t_i}} - \frac{\alpha_{t_{i-1}}}{\sigma_{t_{i-1}}}\right) \cdot \boldsymbol{x}_{\boldsymbol{\theta}}(\tilde{\boldsymbol{x}}_{t_{i-1}}, t_{i-1}), \tag{11}$$

which is in line with the widely-used DDIM sampling method (Song et al., 2021a).

Moreover, second- or third-order numerical schemes can be employed to enhance the sampling efficiency further (Lu et al., 2022b; Zhao et al., 2024). Specifically, the second-order multi-step numerical solver for Eq. (10) yields the following iterative formula for $i \geq 2$ (Lu et al., 2022b):

$$\tilde{\boldsymbol{x}}_{t_i} = \frac{\sigma_{t_i}}{\sigma_{t_{i-1}}}\tilde{\boldsymbol{x}}_{t_{i-1}} - \alpha_{t_i}\left(e^{-h_i} - 1\right)$$
$$\times \left(\left(1 + \frac{1}{2r_i}\right)\boldsymbol{x}_{\boldsymbol{\theta}}(\tilde{\boldsymbol{x}}_{t_{i-1}}, t_{i-1}) - \frac{1}{2r_i}\boldsymbol{x}_{\boldsymbol{\theta}}(\tilde{\boldsymbol{x}}_{t_{i-2}}, t_{i-2})\right), \tag{12}$$

where $h_i = \lambda_{t_i} - \lambda_{t_{i-1}}$ and $r_i = h_{i-1}/h_i$. We refer to the corresponding sampling method as DM2M for brevity.

---

[1]In practice, it is common to set the end time of sampling as a small $\epsilon > 0$ instead of 0 to avoid numerical instability, as described in (Lu et al., 2022a, Appendix D.1).

Let $\kappa_i$ be the function corresponding to the $i$-th sampling step. For instance, for the DDIM sampling method in Eq. (11),[2] we have

$$\kappa_i(\boldsymbol{x}) := \frac{\sigma_{t_i}}{\sigma_{t_{i-1}}}\boldsymbol{x} + \sigma_{t_i}\left(\frac{\alpha_{t_i}}{\sigma_{t_i}} - \frac{\alpha_{t_{i-1}}}{\sigma_{t_{i-1}}}\right)\cdot\boldsymbol{x_\theta}(\boldsymbol{x}, t_{i-1}). \tag{13}$$

Then, the entire sampling process can be expressed as $\tilde{\boldsymbol{x}}_\epsilon = G(\tilde{\boldsymbol{x}}_T)$, where the generator $G : \mathbb{R}^n \to \mathbb{R}^n$ is the composition of of $\kappa_1, \kappa_2, \ldots, \kappa_N$, i.e.,

$$G(\boldsymbol{x}) = \kappa_N \circ \cdots \circ \kappa_2 \circ \kappa_1(\boldsymbol{x}). \tag{14}$$

In addition, for convenience, for any $t \in (\epsilon, T]$, we use $G_t$ to denote the sampling process from $t$ to $\epsilon$ (and thus $G_T = G$). Specifically, if letting $i_t = \max\{i \in [N] : t_{i-1} \geq t\}$, we have that $G_t$ is the composition of $\kappa_{i_t}, \kappa_{i_t+1}, \ldots, \kappa_N$, i.e.,

$$G_t(\boldsymbol{x}) = \kappa_N \circ \cdots \circ \kappa_{i_t+1} \circ \kappa_{i_t}(\boldsymbol{x}). \tag{15}$$

### 2.2. The Inversion of Diffusion Models

The inversion in DMs retraces the ODE sampling process with the aim of identifying the specific noise vector that generates a given image vector. This inversion is a crucial aspect in various applications, including image editing (Hertz et al., 2023; Kim et al., 2022; Wallace et al., 2023; Patashnik et al., 2023), watermark detection (Wen et al., 2023), and image-to-image translation (Kawar et al., 2022; Su et al., 2023). Recently, the inversion of DMs has attracted increasing attention, and several relevant methods have emerged (Pan et al., 2023; Zhang et al., 2024b; Wallace et al., 2023; Hong et al., 2024; Wang et al., 2024).

For example, the naive DDIM inversion method (Hertz et al., 2023; Kim et al., 2022) has the following iterative procedure:[3]

$$\hat{\boldsymbol{x}}_{t_{i-1}} = \frac{\sigma_{t_{i-1}}}{\sigma_{t_i}}\hat{\boldsymbol{x}}_{t_i} + \sigma_{t_{i-1}}\left(\frac{\alpha_{t_{i-1}}}{\sigma_{t_{i-1}}} - \frac{\alpha_{t_i}}{\sigma_{t_i}}\right)\boldsymbol{x_\theta}(\hat{\boldsymbol{x}}_{t_i}, t_{i-1}). \tag{16}$$

The naive DDIM inversion method circumvents the computational overhead of the implicit numerical scheme by substituting $\boldsymbol{x_\theta}(\hat{\boldsymbol{x}}_{t_{i-1}}, t_{i-1})$ with $\boldsymbol{x_\theta}(\hat{\boldsymbol{x}}_{t_i}, t_{i-1})$. It has been widely used in imaging applications like image editing. Alternatively, by taking into account the first-order numerical discretization of Eq. (10) from $t = \epsilon$ to $t = T$, it is straightforward to derive an iterative formula that is slightly different from Eq. (16):

$$\hat{\boldsymbol{x}}_{t_{i-1}} = \frac{\sigma_{t_{i-1}}}{\sigma_{t_i}}\hat{\boldsymbol{x}}_{t_i} + \sigma_{t_{i-1}}\left(\frac{\alpha_{t_{i-1}}}{\sigma_{t_{i-1}}} - \frac{\alpha_{t_i}}{\sigma_{t_i}}\right)\boldsymbol{x_\theta}(\hat{\boldsymbol{x}}_{t_i}, t_i). \tag{17}$$

Furthermore, similar to DM2M in Eq. (12), by considering the second-order numerical discretization of Eq. (10) from $t = \epsilon$ to $t = T$, we can obtain the corresponding iterative formula for the inversion of DMs.

Letting $v_i$ be the function corresponding to the $i$-th inversion step. Then, the entire inversion procedure can be written as $\hat{\boldsymbol{x}}_T = G^\dagger(\hat{\boldsymbol{x}}_\epsilon)$, where the inversion operator $G^\dagger : \mathbb{R}^n \to \mathbb{R}^n$ is the composition of $v_1, v_2, \ldots, v_N$, i.e.,

$$G^\dagger(\boldsymbol{x}) = v_1 \circ v_2 \circ \cdots \circ v_N(\boldsymbol{x}). \tag{18}$$

In addition, for convenience, for any $t \in [\epsilon, T)$, we use $G_t^\dagger$ to denote the inversion from $t$ to $T$ (and thus $G_\epsilon^\dagger = G^\dagger$). Specifically, if letting $j_t = \min\{j \in [M] : t_j \leq t\}$, we have that $G_t^\dagger$ is the composition of of $v_{j_t}, v_{j_t-1}, \ldots, v_1$, i.e.,

$$G_t^\dagger(\boldsymbol{x}) = v_1 \circ v_2 \circ \cdots \circ v_{j_t}(\boldsymbol{x}). \tag{19}$$

## 3. Setup and Approaches

Recall that the nonlinear observation vector $\boldsymbol{y} \in \mathbb{R}^m$ is assumed to be generated in accordance with the SIM in Eq. (2), where $\boldsymbol{x}^* \in \mathcal{S}^{n-1}$ is the signal to be estimated. We follow (Liu & Liu, 2022) to assume that the *unknown* link function $f$ satisfies the following conditions:

$$\mu := \mathbb{E}_{\boldsymbol{a}\sim\mathcal{N}(\mathbf{0},\mathbf{I}_n)}\left[f\left(\boldsymbol{a}^T\boldsymbol{x}^*\right)\boldsymbol{a}^T\boldsymbol{x}^*\right] \neq 0, \tag{20}$$

$$\mathbb{E}_{\boldsymbol{a}\sim\mathcal{N}(\mathbf{0},\mathbf{I}_n)}\left[f\left(\boldsymbol{a}^T\boldsymbol{x}^*\right)^4\right] < \infty. \tag{21}$$

The conditions in Eqs. (20) and (21) are mild and hold true for various forms of $f$, such as $f(x) = \text{sign}(x)$ for 1-bit measurements and $f(x) = x^3$ for cubic measurements. However, as noted in prior works such as (Liu & Scarlett, 2020a; Liu & Liu, 2022), the condition in Eq. (20) is not satisfied for phase retrieval models with $f(x) = x^2$ or $f(x) = |x|$ (or their noisy versions).

*Remark* 1. The condition in Eq. (20) is a classic and crucial condition for SIMs. For example, it is (albeit implicitly) assumed in the seminal work (Plan & Vershynin, 2016) and in subsequent research that builds upon it. If this condition fails to hold, specifically when $\mu = 0$, the recovery of $\mu\boldsymbol{x}^*$ as (Plan & Vershynin, 2016) and in our Eq. (24) below becomes meaningless. Additionally, we follow (Liu & Liu, 2022) to assume the condition in Eq. (21), which generalizes the assumption that $f(\boldsymbol{a}^T\boldsymbol{x}^*)$ is sub-Gaussian (which is satisfied by quantized measurement models), and accommodates more general nonlinear measurement models such as cubic measurements with $f(x) = x^3$ and their noisy counterparts.

---

[2]For higher-order sampling methods, we can derive analogous representations for $G$ in Eq. (14) and $G_t$ in Eq. (15) using a recursive formula. An illustration of this process can be found in Lemma 5 within Appendix C.

[3]For brevity, we assume that the sampling and inversion of DMs utilize the same time steps. In practice, they might employ different time steps and have a varying number of steps.

We utilize a generator $G : \mathbb{R}^n \to \mathbb{R}^n$ that corresponds to the sampling process of DMs (*cf.* Eq. (14)) to model the underlying signal. Specifically, we assume that $\boldsymbol{x}^* \in \mathcal{R}(G)$,[4] where $\mathcal{R}(G)$ denotes the range of $G$. This assumption is standard for nonlinear measurement models with generative priors and has also been made in prior works such as (Wei et al., 2019; Liu & Scarlett, 2020a).

Let $G^\dagger : \mathbb{R}^n \to \mathbb{R}^n$ be an operator corresponding to an inversion method for DMs (*cf.* Eq. (18)). The work (Liu & Liu, 2022) sets the estimated vector as $\mathcal{P}_G(\mathbf{A}^T \boldsymbol{y}/m)$, where $\mathcal{P}_G$ denotes the projection operator onto the range of $G$. Since performing the projection step can be time-consuming even for conventional generative models like GANs, the authors of (Raj et al., 2019) use the composition of the pseudoinverse of the generator and the generator itself to approximate the projection operator. Building on these works, to reconstruct the direction of the signal $\boldsymbol{x}^*$ from the knowledge of the measurement matrix $\mathbf{A} \in \mathbb{R}^{m \times n}$ and the observed vector $\boldsymbol{y} \in \mathbb{R}^m$ (despite the *unknown* link $f$), a natural idea is to set the estimated vector $\hat{\boldsymbol{x}}$ as follows:

$$\hat{\boldsymbol{x}} = G \circ G^\dagger \left( \frac{1}{m} \mathbf{A}^T \boldsymbol{y} \right). \tag{22}$$

However, such a simple idea does not fully exploit the relationship between $\frac{1}{m} \mathbf{A}^T \boldsymbol{y}$ and the underlying signal $\boldsymbol{x}^*$, and it leads to unsatisfactory empirical results for diffusion models, as we will see in Section 5. In particular, in the approach corresponding to Eq. (22), the inversion step starts from time $\epsilon$ (for an $\epsilon$ close to 0; see Section 2.1), and this implicitly assumes that $\frac{1}{m} \mathbf{A}^T \boldsymbol{y}$ approximately follows the target data distribution $q_0$. However, based on the following lemmas, we observe that $\frac{1}{m} \mathbf{A}^T \boldsymbol{y}$ should instead be regarded as a noisy version of $\boldsymbol{x}^* \sim q_0$, especially when the number of samples $m$ is relatively small compared to the ambient dimension $n$. First, it is easy to obtain the following lemma.

**Lemma 1.** *If $\boldsymbol{\epsilon} \sim \mathcal{N}(\mathbf{0}, \mathbf{I}_n)$, then we have that the following holds with high probability*[5]

$$\|\boldsymbol{\epsilon}\|_\infty \leq C\sqrt{\log(2n)}, \tag{23}$$

*where $C$ is a sufficiently large positive constant.*

Furthermore, we have the following lemma, which demonstrates that $\frac{1}{m} \mathbf{A}^T \boldsymbol{y}$ is a noisy version of $\mu \boldsymbol{x}^*$ (see Eq. (20) for the definition of $\mu$) and approximately characterizes the noise level. The proofs of Lemmas 1 and 2 are placed in Appendix B.

[4]This assumption can also be approximately expressed as $\boldsymbol{x}^* \sim q_0$ when $G$ corresponds to an excellent diffusion modeling of the target data distribution $q_0$. The assumption that $\boldsymbol{x}^* \sim q_0$ is typically implicitly assumed by works that utilize diffusion models to solve compressed sensing problems.

[5]Here and in the rest of the paper, a statement is said to hold with high probability (w.h.p.) if it holds with probability at least 0.99.

**Lemma 2.** *Under conditions for the link function $f$ in Eqs. (20) and (21), we have the following holds w.h.p.:*

$$\left\| \frac{1}{m} \mathbf{A}^T \boldsymbol{y} - \mu \boldsymbol{x}^* \right\|_\infty \leq \frac{C'\sqrt{\log(2n)}}{\sqrt{m}}, \tag{24}$$

*where $C'$ is a sufficiently large positive constant.*

The comparison between Lemmas 1 and 2 inspires us to express $\frac{1}{m\mu} \mathbf{A}^T \boldsymbol{y}$ as

$$\frac{1}{m\mu} \mathbf{A}^T \boldsymbol{y} \approx \boldsymbol{x}^* + \frac{C'}{C\mu\sqrt{m}} \boldsymbol{\epsilon} \tag{25}$$

for $\boldsymbol{\epsilon} \sim \mathcal{N}(\mathbf{0}, \mathbf{I}_n)$. Additionally, from Eq. (3), we know that $\boldsymbol{x}_t$ can be written as $\boldsymbol{x}_t = \alpha_t \boldsymbol{x}_0 + \sigma_t \boldsymbol{\epsilon}_t = \alpha_t(\boldsymbol{x}_0 + (\sigma_t/\alpha_t)\boldsymbol{\epsilon}_t)$ with $\boldsymbol{\epsilon}_t \sim \mathcal{N}(\mathbf{0}, \mathbf{I}_n)$. Since the nonlinear link function $f$ is *unknown* (and consequently, $\mu$ is also unknown), and $C, C'$ are undetermined positive constants, we employ a tuning parameter $C_s$ and find a time $t^*$ such that[6]

$$\frac{\sigma_{t^*}}{\alpha_{t^*}} = \frac{C_s}{\sqrt{m}}, \tag{26}$$

and start the inversion from time $t^*$ with the input vector being $\alpha_{t^*} C'_s \mathbf{A}^T \boldsymbol{y}/m$, where $C'_s > 0$ is a tuning parameter. *Remark 2.* Unlike prior works such as (Chung et al., 2022b; Fabian et al., 2024; Wu et al., 2024) that identify an intermediate time for sampling, we seek an intermediate time mainly for the inversion of DMs. Moreover, this intermediate time is computed using theoretical findings in Lemma 2.

Then, the corresponding estimated vector is

$$\hat{\boldsymbol{x}} = G \circ G_{t^*}^\dagger \left( \frac{\alpha_{t^*} C'_s}{m} \mathbf{A}^T \boldsymbol{y} \right), \tag{27}$$

where $G_{t^*}^\dagger$ denotes the partial inversion operator (from $t^*$ to $T$; see Eq. (19)). One may also notice that since $\frac{1}{m} \mathbf{A}^T \boldsymbol{y}$ can be regarded as a noisy version of $\boldsymbol{x}^*$ and the sampling process has a denoising effect, we can simply sample from $t^*$ to $\epsilon$ without taking the inversion step. In this case, the corresponding estimated vector is

$$\hat{\boldsymbol{x}} = G_{t^*} \left( \frac{\alpha_{t^*} C'_s}{m} \mathbf{A}^T \boldsymbol{y} \right), \tag{28}$$

where $G_{t^*}$ denotes the partial sampling operator (from $t^*$ to $\epsilon$; see Eq. (15)). Indeed, such a simple and efficient approach can also lead to reasonably good reconstructions, although its performance is inferior to that of the approach corresponding to Eq. (27).

We refer to the approach for Eq. (22) as SIM-DMFIS (SIMs using diffusion models with full inversion and sampling

[6]If $\frac{C_s}{\sqrt{m}} > \frac{\sigma_T}{\alpha_T}$, set $t^*$ to $T$, and if $\frac{C_s}{\sqrt{m}} < \frac{\sigma_\epsilon}{\alpha_\epsilon}$, set $t^*$ to $\epsilon$.

procedures), the approach for Eq. (28) as SIM-DMS since it only performs a (partial) sampling procedure, and the approach for Eq. (27) as SIM-DMIS. We illustrate the differences among these three approaches in Figure 1. For convenience, we present the SIM-DMIS approach in Algorithm 1.

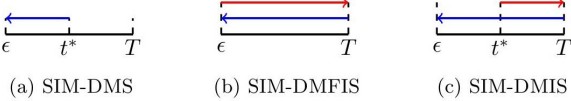

(a) SIM-DMS     (b) SIM-DMFIS     (c) SIM-DMIS

*Figure 1.* An illustration of our three approaches. For SIM-DMS, we only perform the sampling from $t^*$ to $\epsilon$. For SIM-DMFIS, we perform the full inversion and sampling procedures. For SIM-DMIS, we first perform the inversion from $t^*$ to $T$, and then perform the sampling from $T$ to $\epsilon$.

---

**Algorithm 1** The SIM-DMIS approach

---

**Input:** $\mathbf{A} \in \mathbb{R}^{m \times n}$, $\boldsymbol{y} \in \mathbb{R}^m$, $\boldsymbol{x_\theta}$, time steps $\epsilon = t_N < t_{N-1} < \ldots < t_1 < t_0 = T$, generator $G$ corresponding to the sampling process as in Eq. (14), partial inversion operator $G_t^\dagger$ as in Eq. (19), tuning parameters $C_s$ and $C_s'$
**1:** Calculate $t^* \in [\epsilon, T]$ as in Eq. (26).
**2:** Calculate the estimated vector $\hat{\boldsymbol{x}}$ as in Eq. (27).
**Output:** $\hat{\boldsymbol{x}}$

---

## 4. Theoretical Analysis

In this section, we provide a theoretical analysis of the effectiveness of our proposed SIM-DMIS approach. Before presenting the analysis, we first present some preliminary results. Firstly, the following assumption states that the neural function $\boldsymbol{x_\theta}(\boldsymbol{x}, t)$ is Lipschitz continuous with respect to its first argument. This assumption has been widely adopted in relevant works such as (Lu et al., 2022a; Chen et al., 2023a;c; 2024; 2023d; Bortoli, 2022; Lee et al., 2022; Li et al., 2023).

**Assumption 1** (Lipschitz Continuity of the Neural Function). *For any $t \in [0, T]$, there exists a positive constant $L_t > 0$ such that the neural function $\boldsymbol{x_\theta}(\boldsymbol{x}, t)$ is $L_t$-Lipschitz continuous with respect to its first parameter $\boldsymbol{x}$. That is, it holds for all $\boldsymbol{x}_1, \boldsymbol{x}_2 \in \mathbb{R}^n$ that $\|\boldsymbol{x_\theta}(\boldsymbol{x}_1, t) - \boldsymbol{x_\theta}(\boldsymbol{x}_2, t)\|_2 \leq L_t \|\boldsymbol{x}_1 - \boldsymbol{x}_2\|_2$.*

Based on Assumption 1, it can be easily demonstrated that the generator corresponding to popular sampling methods such as DDIM and DM2M (*cf.* Sec. 2.1) is Lipschitz continuous. Specifically, we have the following lemma.

**Lemma 3.** *Suppose that the data prediction network $\boldsymbol{x_\theta}$ satisfies Assumption 1. Then, if $G : \mathbb{R}^n \to \mathbb{R}^n$ is the generator as in Eq. (14), we have that $G$ is $L$-Lipschitz continuous with with $L > 0$ being dependent on $\{\alpha_{t_i}\}_{i=0}^{N-1}$, $\{\sigma_{t_i}\}_{i=0}^{N-1}$, and $\{L_{t_i}\}_{i=0}^{N-1}$.*

Lemma 3 shows that it is reasonable to assume that $G$ is Lipschitz continuous. The proof of Lemma 3 and the precise characterization of the Lipschitz constant $L$ for DDIM and DM2M are deferred to Appendix C.

Then, we have the following theorem regarding the effectiveness of the SIM-DMIS approach as presented in Eq. (27). Specifically, for $t \in [\epsilon, T]$, it offers an upper bound for the distance between $G \circ G_t^\dagger(\bar{\boldsymbol{x}}_t)$ and $\bar{\boldsymbol{x}}_\epsilon$, where $\bar{\boldsymbol{x}}_t$ is a sample of $q_t$ and $\bar{\boldsymbol{x}}_\epsilon$ is on the same ODE trajectory as $\bar{\boldsymbol{x}}_t$. When $\epsilon$ is sufficiently small and the (scaled) data prediction network $\boldsymbol{x_\theta}$ provides a good approximation to the ground-truth score $\nabla \log q_t$ (Chen et al., 2023d; Li et al., 2023), $\bar{\boldsymbol{x}}_\epsilon$ will be close to the ground-truth data following $q_0$ that corresponds to $\bar{\boldsymbol{x}}_t$. Additionally, it has been demonstrated in (Lu et al., 2022a;b) that under certain regularity conditions, DDIM and DM2M are first- and second-order numerical sampling methods for Eq. (10), respectively. Moreover, by considering the first- and second-order numerical discretization of Eq. (10) from $t = \epsilon$ to $t = T$, we similarly obtain the first- and second-order numerical inversion methods for Eq. (10) respectively (for instance, the inversion method in Eq. (17) is a first-order numerical inversion method).

Therefore, based on Theorem 3, if the term $\alpha_{t^*} C_s' \mathbf{A}^T \boldsymbol{y}/m$ in Eq. (27) can be approximately expressed as $\alpha_{t^*} \boldsymbol{x}^* + \sigma_{t^*} \boldsymbol{\epsilon}$ with $\boldsymbol{\epsilon}$ being a standard normal vector, $\hat{\boldsymbol{x}} := G \circ G_{t^*}^\dagger (\alpha_{t^*} C_s' \mathbf{A}^T \boldsymbol{y}/m)$ is close to $\boldsymbol{x}^*$ under appropriate conditions. The proof of Theorem 3 can be found in Appendix C.

**Theorem 3.** *For $t \in [\epsilon, T]$, let $\bar{\boldsymbol{x}}_t \in \mathbb{R}^n$ be a sample of $q_t$, and let $\bar{\boldsymbol{x}}_\epsilon = \frac{\sigma_\epsilon}{\sigma_t} \bar{\boldsymbol{x}}_t + \sigma_\epsilon \int_{\lambda_t}^{\lambda_\epsilon} e^\lambda \hat{\boldsymbol{x}}_\theta(\hat{\boldsymbol{x}}_\lambda, \lambda) \mathrm{d}\lambda$, which is the analytic solution of Eq. (10) with respect to the initial vector $\bar{\boldsymbol{x}}_t$. For $k_1, k_2 \in \{1, 2, 3\}$, suppose that $G^\dagger$ corresponds to a $k_1$-th order numerical inversion method for Eq. (10), and $G_t^\dagger$ denotes the partial inversion operator from $t$ to $T$ (see Eq. (19)). Suppose that the generator $G$ corresponds to a $k_2$-th order numerical sampling method for Eq. (10) and $G$ is $L$-Lipschitz continuous. Then, under certain regularity conditions,[7] we have the following:*

$$\|\bar{\boldsymbol{x}}_\epsilon - G \circ G_t^\dagger(\bar{\boldsymbol{x}}_t)\|_2 = O\left(\sqrt{n}\left(h_{\max}^{k_2} + L h_{\max}^{k_1}\right)\right), \quad (29)$$

*where $h_{\max} = \max_{i \in [N]}(\lambda_{t_i} - \lambda_{t_{i-1}})$.*

## 5. Experiments

In this section, we conduct a series of experiments to validate the effectiveness of the proposed SIM-DMIS approach (see Algorithm 1). Specifically, we evaluate our method on two datasets: FFHQ 256×256 (Karras et al., 2019), ImageNet 256×256 (Deng et al., 2009).[8] For FFHQ and Im-

---

[7]These conditions are similar to those in (Lu et al., 2022a, Appendix B.1) and are listed in Appendix C.

[8]Due to the page limit, the results of CIFAR-10 (Krizhevsky & Hinton, 2009) are included in Appendix D.

ageNet, the ambient dimension is $n = 3 \times 256 \times 256 = 196608$.

We perform experiments on nonlinear measurement models with 1-bit and cubic measurements, for which we have $\boldsymbol{y} = \text{sign}(\mathbf{A}\boldsymbol{x}^* + \boldsymbol{e})$ and $\boldsymbol{y} = (\mathbf{A}\boldsymbol{x}^*)^3 + \boldsymbol{e}$ respectively, where $\boldsymbol{e} \sim \mathcal{N}(\mathbf{0}, \sigma^2 \mathbf{I}_m)$. We compare `SIM-DMIS` with `QCS-SGM` proposed in (Meng & Kabashima, 2022), which uses DMs to handle quantized compressed sensing problems. We also compare our approach with the posterior sampling methods `DPS` (Chung et al., 2023b) and `DAPS` (Zhang et al., 2024a). `DPS` is a popular baseline for general signal recovery problems, and `DAPS` has shown excellent performance in nonlinear signal recovery problems. When `DPS` and `DAPS` utilize the knowledge of the link function $f$,[9] the corresponding methods are denoted as `DPS-N` and `DAPS-N` respectively. In addition, we compare with `DPS` and `DAPS` for the linear operator without using the knowledge of $f$, and the corresponding methods are denoted as `DPS-L` and `DAPS-L` respectively.

For the `QCS-SGM` method, we adhere to the experimental settings specified in (Meng & Kabashima, 2022) to ensure a fair comparison. Additionally, we compare with `SIM-DMFIS` and `SIM-DMS` (see Eqs. (22) and (28)), which are two variants of `SIM-DMIS`. We employ three metrics to assess the performance of these methods: Peak signal-to-noise ratio (PSNR), structural similarity index measure (SSIM), and learned perceptual image patch similarity (LPIPS). The reported quantitative results are averaged over 100 testing images. In the experimental process, we use DDIM for sampling and the DM2M inversion method (refer to Sec. 2.1 for details). The reported NFEs represent the total number of NFEs utilized by each approach. For instance, in the case of `SIM-DMIS` and `SIM-DMFIS`, the reported NFE is the sum of the number of steps in the inversion process and the number of steps in the sampling process. We adopt the variance preserving (VP) schedule with $\beta_{\min} = 0.1$ and $\beta_{\max} = 20$. For `SIM-DMS`, we use 50 NFEs, while for `SIM-DMIS` and `SIM-DMFIS`, we use 150 NFEs.[10] The remaining parameters are set as $\epsilon = 0.001$, $T = 1$, and $\sigma = 0.05$. For the detailed parameter settings of $C_s$ and $C_s'$ in `SIM-DMS` and `SIM-DMIS`, please refer to Appendices F and G.

**Results for FFHQ ($256 \times 256$)** Since the experiments for FFHQ are time-consuming, we only report the results for $m = n/8 = 24576$. For the `QCS-SGM` method, we leverage a pre-trained unconditional Score-SDE model with the

variance exploding (VE) schedule[11] for the FFHQ dataset. As for the other methods, we employ the model from `DPS`, which has been trained on 49,000 FFHQ $256 \times 256$ images and validated on the first 1,000 images.

It is evident from Table 1 and Figure 2 that `SIM-DMIS` exhibits superior reconstruction performance with 1-bit measurements. Notably, `SIM-DMIS` requires only 150 NFEs, in contrast to `QCS-SGM`, which demands a significantly higher 11,555 NFEs. Additionally, both `DPS` and `DAPS` utilize 1,000 NFEs. The results for cubic measurements presented in Table 2 and Figure 3 further demonstrate the effectiveness of our proposed methods. In particular, `SIM-DMIS` consistently achieves the highest reconstruction quality across all evaluated metrics while requiring only 150 NFEs, demonstrating both the efficiency and robustness of our approach under the cubic measurement setting on the FFHQ dataset.

Note that in 1-bit measurements, even in the noiseless scenario, the link function $f(x) = \text{sign}(x)$ is non-differentiable at $x = 0$ (though as mentioned in Footnote 9, Py-Torch can still enforce automatic differentiation). The non-differentiability also poses challenges for `DPS-N` and `DAPS-N` as they rely on $f$ in gradient based updates. This can lead to inaccurate gradients and ultimately resulting in subpar performance.[12] For cubic measurements, since we have observed from the results for 1-bit measurements that `SIM-DMFIS`, `DPS-L`, and `DAPS-L` do not perform well, we do not compare against them. Furthermore, we do not include comparisons with `DAPS-N` since although the link function for cubic measurements is differentiable, its pronounced non-linearity produces unstable gradient directions that make optimization with `DAPS-N` unreliable and significantly degrade reconstruction performance.

*Table 1.* Quantitative results for FFHQ with 1-bit measurements ($m = n/8 = 24576$). We mark **bold** the best scores, and underline the second best scores.

| Method | NFE | PSNR ($\uparrow$) | SSIM ($\uparrow$) | LPIPS ($\downarrow$) |
|---|---|---|---|---|
| QCS-SGM | 11555 | 12.91±2.36 | 0.51±0.07 | 0.50±0.08 |
| DPS-N | 1000 | 11.14±1.46 | 0.37±0.09 | 0.69± 0.05 |
| DPS-L | 1000 | 8.57±2.05 | 0.22±0.08 | 0.69±0.09 |
| DAPS-N | 1000 | 16.59±0.54 | 0.33±0.05 | 0.48±0.05 |
| DAPS-L | 1000 | 5.63±0.71 | 0.04±0.03 | 0.61±0.03 |
| SIM-DMS | 50 | 17.14±2.41 | 0.44±0.07 | 0.48±0.05 |
| SIM-DMFIS | 150 | 8.48±0.13 | 0.03±0.00 | 0.90±0.02 |
| SIM-DMIS | 150 | **19.87**±2.77 | **0.60**±0.09 | **0.37**±0.05 |

---

[9] For 1-bit measurements where $f$ is not always differentiable, PyTorch can still enforce automatic differentiation.

[10] The same number of NFEs is used for the inversion process of `SIM-DMIS` and `SIM-DMFIS`. As demonstrated in Appendices F and G, this choice has a negligible impact on the experimental results.

[11] https://huggingface.co/google/ncsnpp-ffhq-256

[12] The supplementary material of DAPS suggests using Metropolis Hasting for non-differentiable forward operators, but it is also mentioned to have inferior performance and low efficiency, with results only in the supplementary material.

*Table 2.* Quantitative results on the FFHQ with cubic measurements ($m = n/8 = 24576$). We mark **bold** the best scores, and underline the second best scores.

| Method | NFE | PSNR ($\uparrow$) | SSIM ($\uparrow$) | LPIPS ($\downarrow$) |
|---|---|---|---|---|
| DPS-N | 1000 | 15.41±1.38 | 0.37±0.08 | 0.50±0.07 |
| SIM-DMS | 50 | 14.44±2.13 | 0.51±0.10 | 0.47±0.08 |
| SIM-DMIS | 150 | **17.56**±2.56 | **0.57**±0.10 | **0.40**±0.07 |

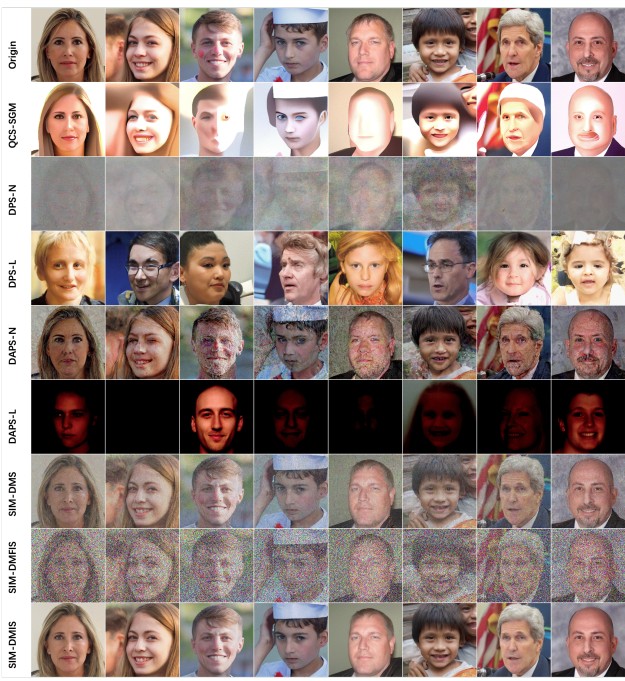

*Figure 2.* Examples of 1-bit reconstructed images for FFHQ.

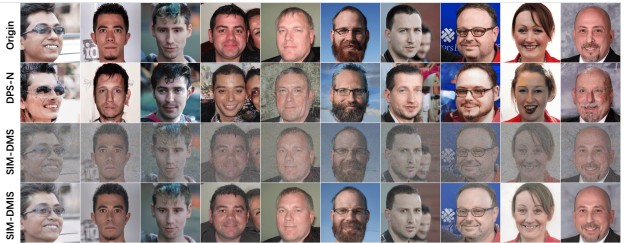

*Figure 3.* Examples of cubic reconstructed images for FFHQ.

**Results for ImageNet (256×256)**    For ImageNet, we report the results for $m = n/16 = 12288$. We utilize a pretrained conditional ImageNet 256×256 model along with its corresponding classifier sourced from ADM (Dhariwal & Nichol, 2021). We adhere to the default recommended configuration settings of these models, substituting the unconditional ImageNet 256×256 model initially provided by DPS.[13] The tests are conducted on 100 images selected from the Validation1K Set.[14] Given that QCS-SGM lacks pre-trained models for ImageNet, we do not compare with it.

Table 3 and Figure 4 demonstrate that SIM-DMIS achieves superior reconstruction with 1-bit measurements compared to DPS and DAPS variants, while requiring only 150 NFEs. The results with cubic measurements, summarized in Table 4 and Figure 5, further validate the advantages of our methods. Notably, SIM-DMIS achieves the best performance across all metrics while maintaining a low computational cost of only 150 NFEs. These results underscore the effectiveness and efficiency of our approach in handling cubic measurement settings on the ImageNet dataset.

*Table 3.* Quantitative results for ImageNet with 1-bit measurements ($m = n/16 = 12288$). We mark **bold** the best scores, and underline the second best scores.

| Method | NFE | PSNR ($\uparrow$) | SSIM ($\uparrow$) | LPIPS ($\downarrow$) |
|---|---|---|---|---|
| DPS-N | 1000 | 11.57±2.63 | 0.24±0.11 | 0.64±0.04 |
| DPS-L | 1000 | 12.67±3.43 | 0.11±0.16 | 0.81±0.12 |
| DAPS-N | 1000 | 14.62±1.20 | 0.12±0.01 | 0.61±0.04 |
| DAPS-L | 1000 | 6.20±1.66 | 0.08±0.14 | 0.64±0.02 |
| SIM-DMS | 50 | 16.93±3.16 | 0.32±0.09 | 0.54±0.09 |
| SIM-DMFIS | 150 | 11.89±2.24 | 0.08±0.02 | 0.76±0.06 |
| SIM-DMIS | 150 | **18.06**±3.16 | **0.45**±0.13 | **0.46**±0.08 |

*Table 4.* Quantitative results on the ImageNet with cubic measurements ($m = n/16 = 12288$). We mark **bold** the best scores, and underline the second best scores.

| Method | NFE | PSNR ($\uparrow$) | SSIM ($\uparrow$) | LPIPS ($\downarrow$) |
|---|---|---|---|---|
| DPS-N | 1000 | 11.47±2.97 | 0.22±0.08 | 0.62±0.06 |
| SIM-DMS | 50 | 14.99±2.82 | 0.33±0.06 | 0.59±0.08 |
| SIM-DMIS | 150 | **16.28**±3.38 | **0.40**±0.15 | **0.55**±0.10 |

---

[13]For DAPS, we follow the settings described in the original paper and employ a pre-trained unconditional model from ADM (Dhariwal & Nichol, 2021).

[14]https://github.com/XingangPan/deep-generative-prior/

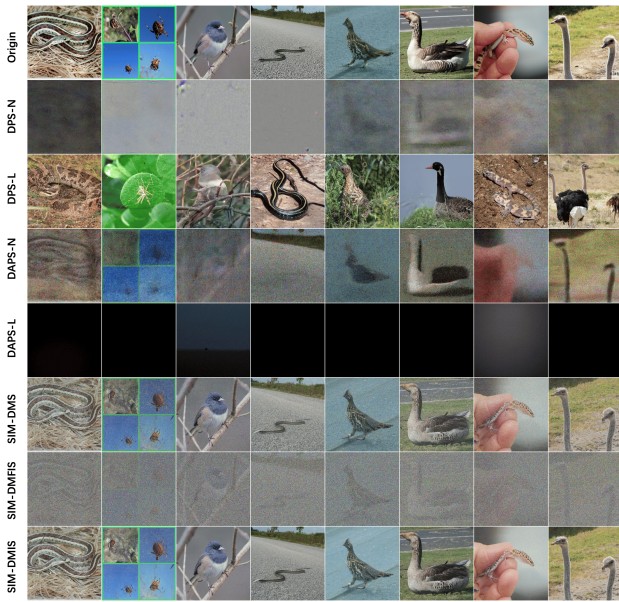

*Figure 4.* Examples of 1-bit reconstructed images for ImageNet.

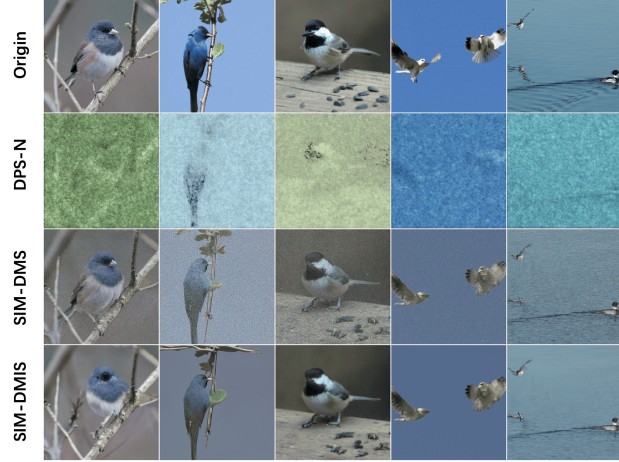

*Figure 5.* Examples of cubic reconstructed images for ImageNet.

## 6. Conclusion

In this paper, we present approaches for learning SIMs by making use of the sampling and inversion methods for pre-trained unconditional DMs. Theoretical analysis and numerical results are provided to illustrate the effectiveness of the proposed approaches.

## Impact Statement

This paper presents work aimed at advancing the field of Machine Learning, particularly in the area of signal recovery using diffusion models (DMs). The proposed method applies DMs to enhance the recovery of signals from Single Index Models (SIMs), offering more flexible and robust solutions for complex signal reconstruction tasks. It could have broad applications in fields such as imaging, signal processing, and data reconstruction, with potential to improve the efficiency and accuracy of various real-world tasks. While no direct ethical concerns are associated with the methodology, the potential societal impacts of this work include improvements in computational efficiency and the enhancement of technologies reliant on image reconstruction and signal recovery.

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

# A. Extended Related Work

In this subsection, we summarize some relevant works, which can roughly be divided into (i) nonlinear signal recovery with conventional generative models, and (ii) signal recovery with diffusion models.

**Nonlinear signal recovery with conventional generative models:**    The seminal work (Bora et al., 2017) studies linear compressed sensing with conventional generative priors (such as GANs and VAEs) and demonstrates via numerical experiments on image datasets that using a pre-trained generative prior can significantly reduce the number of required measurements (compared to that of using the sparse prior) for accurate signal recovery. This has led to a significant volume of follow-up works in (Dhar et al., 2018; Hand et al., 2018; Shah & Hegde, 2018; Jagatap & Hegde, 2019; Asim et al., 2020; Ongie et al., 2020; Daras et al., 2021; Liu & Scarlett, 2020b; Jalal et al., 2021b;a), which explore various aspects of high-dimensional signal recovery with generative priors. A literature review in this area can be found in (Scarlett et al., 2022).

In particular, under conventional generative priors, there have been several studies on 1-bit compressed sensing (Qiu et al., 2020; Liu et al., 2020; 2021a) and phase retrieval (Hand et al., 2018; Hyder et al., 2019; Jagatap & Hegde, 2019; Aubin et al., 2020; Shamshad & Ahmed, 2020; Liu et al., 2021b; Killedar & Seelamantula, 2022; Liu et al., 2022b), among others. Moreover, SIMs that account for unknown nonlinearity and conventional generative priors have been explored in several studies, including (Wei et al., 2019; Liu & Scarlett, 2020a; Liu & Liu, 2022; Chen et al., 2023b). More specifically, the works (Liu & Scarlett, 2020a; Liu & Liu, 2022; Chen et al., 2023b) provide recovery guarantees for the generalized Lasso approach and practical algorithms under the assumption of Gaussian sensing vectors. In (Wei et al., 2019), under the extra assumption that the link function is differentiable, SIMs with link functions of first- and second-orders are studied through Stein's identity, allowing for general non-Gaussian sensing vectors.

**Signal recovery with diffusion models:**    Due to the ability to model complex distributions, DMs have demonstrated remarkable performance in a variety of applications. The seminal work on DMs (Sohl-Dickstein et al., 2015) introduces a framework for the diffusion process, which has been further developed in multiple forms, such as denoising diffusion probabilistic models (DDPMs) (Ho et al., 2020) and score-based generative models (SGMs) (Song et al., 2021b). These models have been widely applied to signal recovery problems (Daras et al., 2024), which can be broadly classified into two main paradigms.

One paradigm is task-specific, in which DMs are trained for particular tasks. For example, the model in (Saharia et al., 2022c) is customized for super-resolution, while the model in (Ren et al., 2023) is developed for deblurring. The other paradigm makes use of pre-trained DMs that are not limited to any specific signal recovery problem. Works within this paradigm can be further classified according to how they handle the measurements. Some approaches use DMs as priors to direct the reconstruction process by estimating the score function, as shown in (Feng et al., 2023; Daras et al., 2022). Other approaches, such as DDRM (Kawar et al., 2022), adapt the DDPM framework specifically for linear signal recovery problems. By incorporating singular value decomposition to limit the solution space, DDRM can effectively handle linear measurements. However, its performance degrades when dealing with sparse or limited measurements, rendering it less effective in challenging situations.

To solve general noisy linear signal recovery problems, some recent works (Chung et al., 2022a; 2023b; Song et al., 2023; Fei et al., 2023; Fabian et al., 2023; Zhang et al., 2023) aim to estimate the posterior distribution using the unconditional diffusion model based on Bayes' rule. In particular, MCG (Chung et al., 2022a) approaches data consistency from the perspective of the data manifold. It proposes a manifold-constrained gradient to ensure that corrections stay on the data manifold. DPS (Chung et al., 2023b) abandons the projection step in the reverse process because it may cause the sampling path to deviate from the data manifold. Instead, it approximates the gradient of the posterior through a specially hand-designed strategy, where the measurements can be regarded as a signal conditioning the sampling process. ΠGDM (Song et al., 2023) further develops this type of strategy into a unified expression that encompasses various signal recovery problems by using the Moore-Penrose pseudoinverse. ΠGDM guides the diffusion process by matching the one-step denoising solution and the ground-truth measurements, after transforming both via a generalized pseudoinverse of the measurement model. DAPS (Zhang et al., 2024a) modifies the traditional diffusion process by decoupling consecutive steps in the sampling trajectory. This allows for greater variation between steps, improving the exploration of the solution space and leading to more accurate and stable reconstructions.

# B. Proofs of Lemmas 1 and 2

Before presenting the proofs, we present the following simple tail bound for a Gaussian random variable.

**Lemma 4.** (Gaussian tail bound (Wainwright, 2019, Example 2.1)) *Suppose that $X \sim \mathcal{N}(\alpha, \sigma^2)$ is a Gaussian random variable with mean $\alpha$ and variance $\sigma^2$. Then, for any $u > 0$, $\mathbb{P}(|X - \alpha| \geq u) \leq 2e^{-\frac{u^2}{2\sigma^2}}$.*

## B.1. Proof of Lemma 1

For $u > 0$ and any $j \in [n]$, from Lemma 4, from Lemma 4, we have with probability at least $1 - 2e^{-\frac{u^2}{2}}$ that $|\epsilon_j| \leq u$. Taking a union bound over all $j \in [n]$, we obtain with probability at least $1 - 2ne^{-\frac{u^2}{2}}$ that $\|\boldsymbol{\epsilon}\|_\infty \leq u$. Then, by setting $u = C\sqrt{\log(2n)}$, we obtain the desired result.

## B.2. Proof of Lemma 2

Let $g_i = \boldsymbol{a}_i^T \boldsymbol{x}^*$. Since $\boldsymbol{x}^* \in \mathcal{S}^{n-1}$, we know that $g_i$ are independent standard normal random variables. Additionally, note that

$$\frac{1}{m}\mathbf{A}^T \boldsymbol{y} - \mu \boldsymbol{x}^* = \frac{1}{m}\sum_{i=1}^m y_i \boldsymbol{a}_i - \mu \boldsymbol{x}^* \tag{30}$$

$$= \frac{1}{m}\sum_{i=1}^m y_i (\boldsymbol{a}_i - (\boldsymbol{a}_i^T \boldsymbol{x}^*)\boldsymbol{x}^*) + \frac{1}{m}\sum_{i=1}^m y_i (\boldsymbol{a}_i^T \boldsymbol{x}^*)\boldsymbol{x}^* - \mu \boldsymbol{x}^*. \tag{31}$$

For any $j \in [n]$, since $\mathbb{E}[a_{ij}g_i] = x_j^*$, $a_{ij}$ can be written as

$$a_{ij} = x_j^* g_i + \sqrt{1 - (x_j^*)^2}\, r_{ij}, \tag{32}$$

where $r_{ij}$ is a standard normal random variable that is independent of $g_i$. Then, the $j$-th entry of the first term in the right-hand side of Eq. (31) can be written as

$$\left[\frac{1}{m}\sum_{i=1}^m y_i (\boldsymbol{a}_i - (\boldsymbol{a}_i^T \boldsymbol{x}^*)\boldsymbol{x}^*)\right]_j = \frac{1}{m}\sum_{i=1}^m y_i (a_{ij} - g_i x_j^*) \tag{33}$$

$$= \sqrt{1 - (x_j^*)^2} \cdot \frac{1}{m}\sum_{i=1}^m y_i r_{ij}. \tag{34}$$

Note that $y_i = f(\boldsymbol{a}_i^T \boldsymbol{x}^*) = f(g_i)$. Thus $r_{ij}$ is also independent of $y_i$. Let $M_2 := \mathbb{E}_{g \sim \mathcal{N}(0,1)}[f(g)^2]$ and $\mathcal{E}_1$ be the event that $\frac{1}{m}\sum_{i=1}^m y_i^2 \leq 2M_2$. From (Liu & Liu, 2022, Lemma 3), we have

$$\mathbb{P}(\mathcal{E}_1^c) \leq \frac{C_f}{m}, \tag{35}$$

where $C_f$ is a positive constant depending on $f$. Moreover, conditioned on the event $\mathcal{E}_1$, the term $\sum_{i=1}^m y_i r_{ij}$ in Eq. (34) is zero-mean Gaussian with the variance being $\sum_{i=1}^m y_i^2$. Then, from Lemma 4, we obtain

$$\mathbb{P}\left(\left|\sum_{i=1}^m y_i r_{ij}\right| > u\right) \leq 2\exp\left(-\frac{u^2}{2\sum_{i=1}^m y_i^2}\right) \leq 2\exp\left(-\frac{u^2}{4mM_2}\right). \tag{36}$$

Taking a union bound over all $j \in [n]$, we obtain with probability at least $1 - 2n\exp\left(-\frac{u^2}{4mM_2}\right)$ that

$$\left\|\frac{1}{m}\sum_{i=1}^m y_i (\boldsymbol{a}_i - (\boldsymbol{a}_i^T \boldsymbol{x}^*)\boldsymbol{x}^*)\right\|_\infty = \left\|\sqrt{1 - (x_j^*)^2} \cdot \frac{1}{m}\sum_{i=1}^m y_i r_{ij}\right\|_\infty \leq \frac{u}{m}. \tag{37}$$

Furthermore, the $j$-th entry of the second term in the right-hand side of Eq. (31) can be written as

$$\frac{1}{m}\sum_{i=1}^m y_i (\boldsymbol{a}_i^T \boldsymbol{x}^*)\boldsymbol{x}^* - \mu \boldsymbol{x}^* = \left(\frac{1}{m}\sum_{i=1}^m y_i (\boldsymbol{a}_i^T \boldsymbol{x}^*) - \mu\right)\boldsymbol{x}^*. \tag{38}$$

For any $\tau > 0$, let $\mathcal{E}_2$ be the event that $\left|\frac{1}{m}\sum_{i=1}^m y_i(\boldsymbol{a}_i^T\boldsymbol{x}^*) - \mu\right| < \tau$. From (Liu & Liu, 2022, Lemma 3), we have

$$\mathbb{P}(\mathcal{E}_2^c) \leq \frac{C_f'}{m\tau^2}, \tag{39}$$

where $C_f'$ is a positive constant depending on $f$. Additionally, conditioned on $\mathcal{E}_2$, we have

$$\left\|\left(\frac{1}{m}\sum_{i=1}^m y_i(\boldsymbol{a}_i^T\boldsymbol{x}^*) - \mu\right)\boldsymbol{x}^*\right\|_\infty = \left|\frac{1}{m}\sum_{i=1}^m y_i(\boldsymbol{a}_i^T\boldsymbol{x}^*) - \mu\right| \cdot \|\boldsymbol{x}^*\|_\infty \leq \tau. \tag{40}$$

Combining Eqs. (31), (35), (37), (39), and (40), we observe that by setting $u = C_1\sqrt{\log(2n)} \cdot \sqrt{m}$ and $\tau = C_2\sqrt{\log(2n)}/\sqrt{m}$ for some sufficiently large positive constants $C_1$ and $C_2$ and letting $C' = C_1 + C_2$, we obtain the desired result.

## C. Proof of Theorem 3

First, we provide a proof for the auxiliary result in Lemma 3 for popular sampling methods such as DDIM and DM2M (*cf.* Sec. 2.1). Specifically, we have the following detailed version of Lemma 3.

**Lemma 5.** *Suppose that the data prediction network $\boldsymbol{x_\theta}$ satisfies Assumption 1. Then, if $G : \mathbb{R}^n \to \mathbb{R}^n$ is the generator corresponding to the entire sampling process of DDIM (see Eq. (11)), we have that $G$ is $L$-Lipschitz continuous with*

$$L = \prod_{i=1}^N \left(\frac{\sigma_{t_i}}{\sigma_{t_{i-1}}} + \sigma_{t_i}\left(\frac{\alpha_{t_i}}{\sigma_{t_i}} - \frac{\alpha_{t_{i-1}}}{\sigma_{t_{i-1}}}\right) \cdot L_{t_{i-1}}\right). \tag{41}$$

*Additionally, suppose that $G : \mathbb{R}^n \to \mathbb{R}^n$ is the generator corresponding to the entire sampling process of DM2M (see Eq. (12)), and suppose that for $i = 0, 1, 2, \ldots, N$, $\tilde{\boldsymbol{x}}_{t_i}$ is $\tilde{L}_i$-Lipschitz continous with respect to $\tilde{\boldsymbol{x}}_{t_0}$. Then, we have that $\tilde{L}_0$ can be set to 1 and $\tilde{L}_1$ can be set to be $\frac{\sigma_{t_1}}{\sigma_{t_0}} + \sigma_{t_1}\left(\frac{\alpha_{t_1}}{\sigma_{t_1}} - \frac{\alpha_{t_0}}{\sigma_{t_0}}\right) \cdot L_{t_0}$, and for $i \geq 2$, $\tilde{L}_i$ can be calculated via the following recursive formula:*

$$\tilde{L}_i = \left(\frac{\sigma_{t_i}}{\sigma_{t_{i-1}}} + \alpha_{t_i}\left(1 - e^{-h_i}\right) \cdot \left(1 + \frac{1}{2r_i}\right)L_{t_{i-1}}\right)\tilde{L}_{i-1} + \alpha_{t_i}\left(1 - e^{-h_i}\right) \cdot \frac{1}{2r_i}L_{t_{i-2}}\tilde{L}_{i-2}. \tag{42}$$

*In particular, we have that $G$ is $L$-Lipschitz continuous with $L = \tilde{L}_N$.*

*Proof.* For DDIM, by the triangle inequality, we have the following inequality for any $i \in [N]$ and any $\boldsymbol{x}_1, \boldsymbol{x}_2 \in \mathbb{R}^n$:

$$\|\kappa_i(\boldsymbol{x}_1) - \kappa_i(\boldsymbol{x}_2)\|_2 \leq \left(\frac{\sigma_{t_i}}{\sigma_{t_{i-1}}} + \sigma_{t_i}\left(\frac{\alpha_{t_i}}{\sigma_{t_i}} - \frac{\alpha_{t_{i-1}}}{\sigma_{t_{i-1}}}\right) \cdot L_{t_{i-1}}\right) \cdot \|\boldsymbol{x}_1 - \boldsymbol{x}_2\|_2, \tag{43}$$

where $\kappa_i$ is defined in Eq. (13). Additionally, it is easy to show that if function $f$ is $L_f$-Lipschitz and $g$ is $L_g$-Lipschitz, then their composition $f \circ g$ is $(L_f L_g)$-Lipschitz (Bora et al., 2017). Then, we obtain that the generator $G = \kappa_N \circ \cdots \circ \kappa_2 \circ \kappa_1$ (see Eq. (14)) is $L$-Lipschitz continuous, with $L$ given by Eq. (41). Note that for the first sampling step of DM2M, we need to use the first-order numerical scheme—DDIM. Then, similarly to the case of DDIM, based on Eq. (12), we can obtain the desired recursive formula for the Lipschitz constants. $\square$

Then, we list the regularity conditions on $\boldsymbol{x_\theta}$ required for Theorem 3 as follows. These conditions are similar to those in (Lu et al., 2022a, Appendix B.1).

- The total derivatives $\frac{d^j\hat{\boldsymbol{x}}_\theta(\hat{\boldsymbol{x}}_\lambda, \lambda)}{d\lambda^j}$ (as a function of $\lambda$) exist and are continuous for $j = 0, 1, 2, 3$.

- The neural function $\boldsymbol{x_\theta}(\boldsymbol{x}, t)$ is Lipschitz continuous with w.r.t. its first parameter $\boldsymbol{x}$ (i.e., Assumption 1).

- $h_{\max} = \max_{i \in [N]}(\lambda_{t_i} - \lambda_{t_{i-1}}) = O(1/N)$.

Based on the above auxiliary results, we are now ready to present the proof of Theorem 3.

## C.1. Proof of Theorem 3

Let $\bar{G}_t^\dagger$ be the analytic inversion operator (corresponding to Eq. (10)) from time $t$ to $T$. That is, for any $\boldsymbol{x} \in \mathbb{R}^n$, we have

$$\bar{G}_t^\dagger(\boldsymbol{x}) = \frac{\sigma_T}{\sigma_t}\boldsymbol{x} + \sigma_T \int_{\lambda_t}^{\lambda_T} e^\lambda \hat{\boldsymbol{x}}_{\boldsymbol{\theta}}(\hat{\boldsymbol{x}}_\lambda, \lambda)\mathrm{d}\lambda. \tag{44}$$

Additionally, let $\bar{G}$ be the analytic sampling operator (corresponding to Eq. (10)) from time $T$ to $\epsilon$. That is, for any $\boldsymbol{x} \in \mathbb{R}^n$, we have

$$\bar{G}(\boldsymbol{x}) = \frac{\sigma_\epsilon}{\sigma_T}\boldsymbol{x} + \sigma_\epsilon \int_{\lambda_T}^{\lambda_\epsilon} e^\lambda \hat{\boldsymbol{x}}_{\boldsymbol{\theta}}(\hat{\boldsymbol{x}}_\lambda, \lambda)\mathrm{d}\lambda. \tag{45}$$

Let $\bar{\boldsymbol{x}}_T = \bar{G}_t^\dagger(\bar{\boldsymbol{x}}_t) = \frac{\sigma_T}{\sigma_t}\bar{\boldsymbol{x}}_t + \sigma_T \int_{\lambda_t}^{\lambda_T} e^\lambda \hat{\boldsymbol{x}}_{\boldsymbol{\theta}}(\hat{\boldsymbol{x}}_\lambda, \lambda)\mathrm{d}\lambda$. We have

$$\bar{G} \circ \bar{G}_t^\dagger(\bar{\boldsymbol{x}}_t) = \bar{G}(\bar{\boldsymbol{x}}_T) \tag{46}$$

$$= \frac{\sigma_\epsilon}{\sigma_\epsilon}\bar{\boldsymbol{x}}_T + \sigma_\epsilon \int_{\lambda_T}^{\lambda_\epsilon} e^\lambda \hat{\boldsymbol{x}}_{\boldsymbol{\theta}}(\hat{\boldsymbol{x}}_\lambda, \lambda)\mathrm{d}\lambda \tag{47}$$

$$= \frac{\sigma_\epsilon}{\sigma_T}\left(\frac{\sigma_T}{\sigma_t}\bar{\boldsymbol{x}}_t + \sigma_T \int_{\lambda_t}^{\lambda_T} e^\lambda \hat{\boldsymbol{x}}_{\boldsymbol{\theta}}(\hat{\boldsymbol{x}}_\lambda, \lambda)\mathrm{d}\lambda\right) + \sigma_\epsilon \int_{\lambda_T}^{\lambda_\epsilon} e^\lambda \hat{\boldsymbol{x}}_{\boldsymbol{\theta}}(\hat{\boldsymbol{x}}_\lambda, \lambda)\mathrm{d}\lambda \tag{48}$$

$$= \frac{\sigma_\epsilon}{\sigma_t}\bar{\boldsymbol{x}}_t + \sigma_\epsilon \int_{\lambda_t}^{\lambda_T} e^\lambda \hat{\boldsymbol{x}}_{\boldsymbol{\theta}}(\hat{\boldsymbol{x}}_\lambda, \lambda)\mathrm{d}\lambda + \sigma_\epsilon \int_{\lambda_T}^{\lambda_\epsilon} e^\lambda \hat{\boldsymbol{x}}_{\boldsymbol{\theta}}(\hat{\boldsymbol{x}}_\lambda, \lambda)\mathrm{d}\lambda \tag{49}$$

$$= \frac{\sigma_\epsilon}{\sigma_t}\bar{\boldsymbol{x}}_t + \sigma_\epsilon \int_{\lambda_t}^{\lambda_\epsilon} e^\lambda \hat{\boldsymbol{x}}_{\boldsymbol{\theta}}(\hat{\boldsymbol{x}}_\lambda, \lambda)\mathrm{d}\lambda \tag{50}$$

$$= \bar{\boldsymbol{x}}_\epsilon. \tag{51}$$

Therefore, we obtain

$$\|G \circ G_t^\dagger(\bar{\boldsymbol{x}}_t) - \bar{\boldsymbol{x}}_\epsilon\|_2 = \|G \circ G_t^\dagger(\bar{\boldsymbol{x}}_t) - \bar{G} \circ \bar{G}_t^\dagger(\bar{\boldsymbol{x}}_t)\|_2 \tag{52}$$

$$\leq \|G \circ G_t^\dagger(\bar{\boldsymbol{x}}_t) - G \circ \bar{G}_t^\dagger(\bar{\boldsymbol{x}}_t)\|_2 + \|G \circ \bar{G}_t^\dagger(\bar{\boldsymbol{x}}_t) - \bar{G} \circ \bar{G}_t^\dagger(\bar{\boldsymbol{x}}_t)\|_2 \tag{53}$$

$$\leq L\|G_t^\dagger(\bar{\boldsymbol{x}}_t) - \bar{G}_t^\dagger(\bar{\boldsymbol{x}}_t)\|_2 + \|G(\bar{\boldsymbol{x}}_T) - \bar{G}(\bar{\boldsymbol{x}}_T)\|_2, \tag{54}$$

where the last inequality follows from the $L$-Lipschitz continuity of $G$ and $\bar{\boldsymbol{x}}_T = \bar{G}_t^\dagger(\bar{\boldsymbol{x}}_t)$. Under the regularity conditions listed above, similarly to (Lu et al., 2022a, Theorem 3.2),[15] we have

$$\|G_t^\dagger(\bar{\boldsymbol{x}}_t) - \bar{G}_t^\dagger(\bar{\boldsymbol{x}}_t)\|_2 = O(\sqrt{n}h_{\max}^{k_1}) \tag{55}$$

and

$$\|G(\bar{\boldsymbol{x}}_T) - \bar{G}(\bar{\boldsymbol{x}}_T)\|_2 = O(\sqrt{n}h_{\max}^{k_2}). \tag{56}$$

Combining Eqs. (54), (55), and (56), we obtain the desired result.

# D. Experimental Results for CIFAR-10 with 1-bit and cubic Measurements

In this section, we present the experimental results obtained from the CIFAR-10 dataset (Krizhevsky & Hinton, 2009) using 1-bit and cubic measurements. For the CIFAR-10 dataset, the ambient dimension is $n = 32 \times 32 \times 3 = 3072$, and the pixel values are normalized to the range $[0, 1]$. Prior experimental findings have shown that the `SIM-DMIS` method outperforms `SIM-DMS` and `SIM-DMFIS` in terms of reconstruction performance. Therefore, in this study, we limit our comparison to our `SIM-DMIS` approach against `QCS-SGM`, `DPS-N`, and `DPS-L`. For 1-bit measurements, since `QCS-SGM` cannot handle cubic measurements, we compare against `OneShot` proposed in (Liu & Liu, 2022), which uses GANs to solve nonlinear signal recovery problems. Additionally, since we have observed from the results for 1-bit measurements that `SIM-DMFIS` does not perform well, we do not compare against it. The experimental results demonstrate the superior performance of our proposed method `SIM-DMIS` across different datasets and measurement settings.

---

[15]Note that the term $O(h_{\max}^k)$ in the statement of (Lu et al., 2022a, Theorem 3.2) is added element-wise. Thus when considering the $\ell_2$ norm, we have an extra $\sqrt{n}$ factor, where $n$ is the data dimension.

**Results on CIFAR-10 (32×32) with 1-Bit Measurements** For 1-bit measurements, we report results under two measurement settings: $m = 500$ and $m = 1000$, corresponding to approximately 16% and 32% of the ambient dimension $n$. We utilize a pre-trained unconditional DDPM model with the variance preserving (VP) schedule[16] for CIFAR-10. Prior experimental findings have shown that the SIM-DMIS method outperforms SIM-DMS and SIM-DMFIS in terms of reconstruction performance. Therefore, in this study, we limit our comparison to our SIM-DMIS approach against QCS-SGM, DPS-N, and DPS-L. As shown in Table 5, SIM-DMIS consistently outperforms existing approaches across both $m = 500$ and $m = 1000$. Qualitative results in Figure 6 further confirm its strong reconstruction capability.

*Table 5.* Quantitative results for 1-bit with $m = 500$ and $m = 1000$ measurements. We mark **bold** the best scores, and underline the second best scores.

| Method | NFE | $m = 500$ | | | $m = 1000$ | | |
|---|---|---|---|---|---|---|---|
| | | PSNR (↑) | SSIM (↑) | LPIPS (↓) | PSNR (↑) | SSIM (↑) | LPIPS (↓) |
| QCS-SGM | 1160 | 8.40±3.75 | 0.25±0.19 | 0.55±0.20 | 11.76±4.59 | 0.41±0.22 | 0.41±0.18 |
| DPS-N | 1000 | 9.43±1.89 | 0.04±0.06 | 0.57±0.05 | 9.70±2.19 | 0.04±0.06 | 0.65±0.08 |
| DPS-L | 1000 | 11.53±1.59 | 0.08±0.02 | 0.61±0.03 | 12.24±2.71 | 0.10±0.05 | 0.69±0.07 |
| SIM-DMIS | 150 | **16.24±1.65** | **0.55±0.10** | **0.40±0.10** | **18.61±1.65** | **0.65±0.08** | **0.31±0.07** |

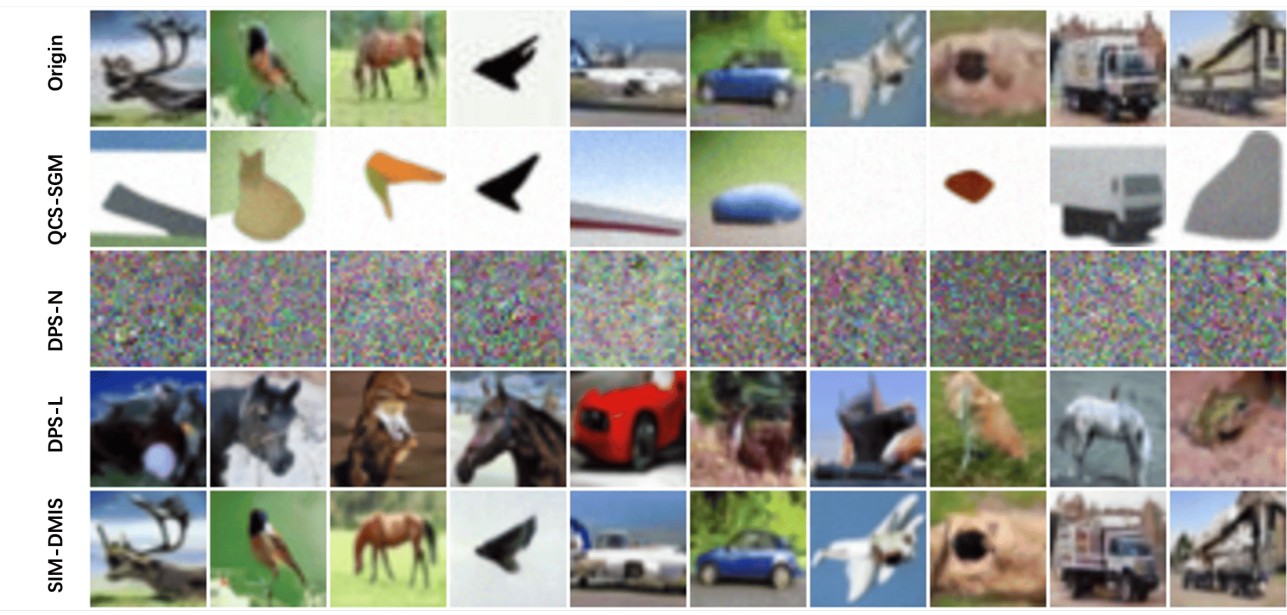

*Figure 6.* Examples of reconstructed images for CIFAR-10 with $m = 1000$ 1-bit measurements.

**Results on CIFAR-10 (32×32) with Cubic Measurements** The quantitative results are shown in Table 6, and examples of reconstructed images are presented in Figure 7. SIM-DMIS consistently outperforms OneShot across all measurement

---

[16]https://huggingface.co/google/ddpm-cifar10-32

settings. Specifically, with $m = 500$ measurements, SIM-DMIS achieves approximately 15% higher PSNR and 2% better SSIM compared to OneShot. With $m = 1000$ measurements, SIM-DMIS achieves approximately 25% higher PSNR and 12% better SSIM compared to OneShot. Furthermore, SIM-DMIS consistently yields a lower LPIPS value, indicating better perceptual similarity to the original images, compared to OneShot.

*Table 6.* Quantitative results for CIFAR-10 with cubic measurements. We mark **bold** the best scores, and underline the second best scores.

| Method | $m = 500$ | | | $m = 1000$ | | |
|---|---|---|---|---|---|---|
| | PSNR (↑) | SSIM (↑) | LPIPS (↓) | PSNR (↑) | SSIM (↑) | LPIPS (↓) |
| OneShot | 13.68±2.09 | 0.33±0.09 | 0.50±0.04 | 14.61±2.09 | 0.39±0.09 | 0.48±0.04 |
| SIM-DMIS | **15.80±2.02** | **0.45±0.10** | **0.43±0.08** | **16.97±2.21** | **0.55±0.12** | **0.38±0.09** |

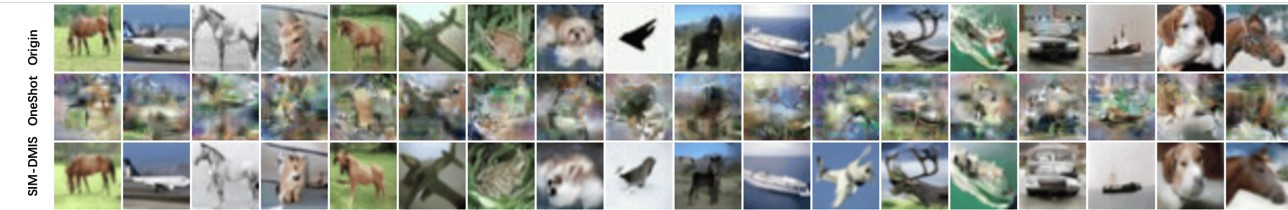

*Figure 7.* Examples of reconstructed images for CIFAR-10 with $m = 1000$ cubic measurements.

# E. Additional Examples of Reconstructed Images for FFHQ and ImageNet with 1-bit Measurements

In this section, we present some additional examples of reconstructed images for the FFHQ 256×256 dataset in Figure 8, and also present some additional examples of reconstructed images for the ImageNet 256×256 dataset in Figure 9.

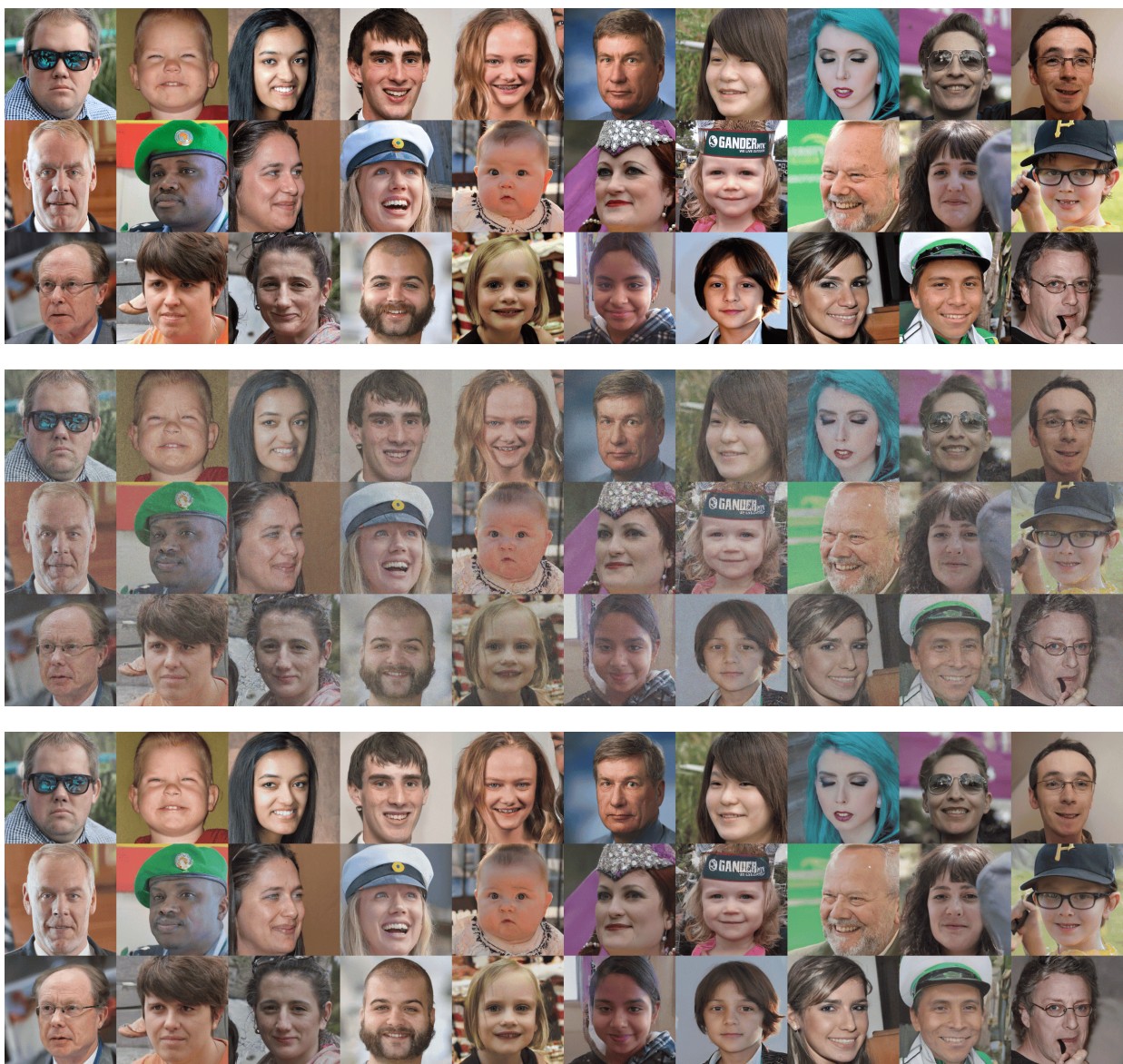

*Figure 8.* Examples of reconstructed images for FFHQ with $m = n/8 = 24576$ 1-bit measurements. Top: Origin, Middle: SIM-DMS, Bottom: SIM-DMIS.

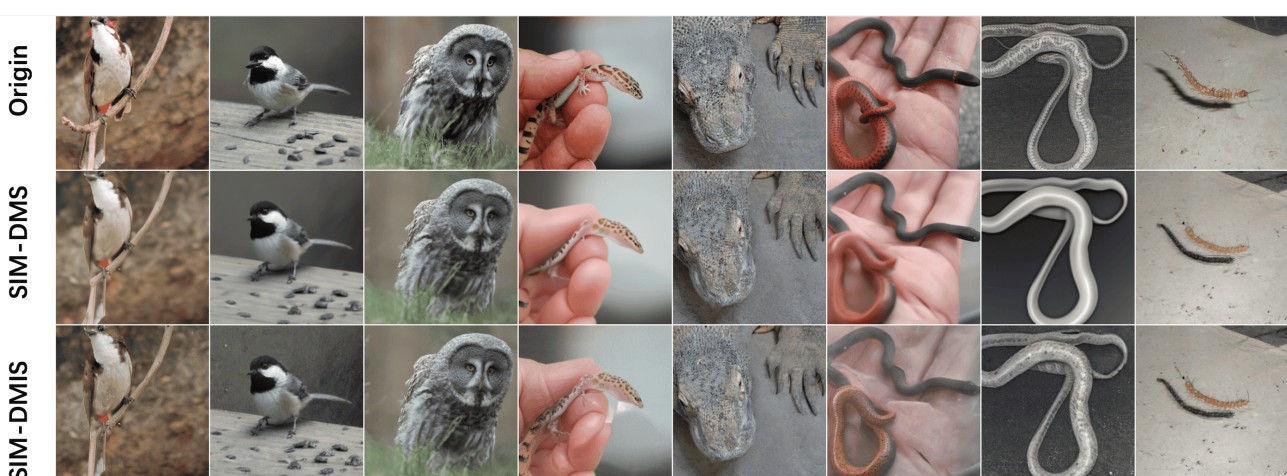

*Figure 9.* Examples of reconstructed images for ImageNet with $m = n/16 = 12288$ 1-bit measurements.

## F. Ablation Study for SIM-DMS

In this section, we present a detailed analysis of the ablation experiments conducted on `SIM-DMS` using the CIFAR-10 dataset with $m = 500$ 1-bit measurements. The experimental setup follows the same framework as in the main body of the research, and the sampling technique is applied throughout.

The results shown in Table 7 provide valuable insights into the performance of `SIM-DMS` under different parameter configurations. Specifically, we analyze the impact of hyperparameters $C_s$, $C'_s$, and the number of function evaluations on model performance.

When varying $C_s$ while fixing $C'_s = 55$ and NFE = 50, we observe a clear upward trend in both PSNR and SSIM, indicating improved signal quality and better structural similarity with increasing $C_s$. However, LPIPS reaches its minimum value at a moderate $C_s$, suggesting that the best perceptual quality is achieved with balanced parameter settings. On the other hand, when $C_s$ is fixed at 55 and $C'_s$ is varied, we find that lower values of $C'_s$ yield better performance across all metrics, with PSNR and SSIM reaching their peak at the lowest tested value. LPIPS remains relatively stable but shows a slight decrease at lower $C'_s$ values. Finally, when we adjust the NFE with $C_s = 55$ and $C'_s = 55$, all performance metrics show minimal fluctuation, suggesting that once NFE reaches a certain threshold, increasing it further does not yield significant improvements.

In conclusion, these results underscore the flexibility of `SIM-DMS` with respect to parameter tuning. The parameter $C_s$ demonstrates a clear impact on reconstruction quality, while lower values of $C'_s$ tend to yield better results. Additionally, the stability across different NFE values indicates the model's efficiency even with fewer iterations. These findings provide valuable guidance for parameter selection in practical applications of `SIM-DMS`.

*Table 7.* Ablation experiments for `SIM-DMS` on CIFAR-10 with $m = 500$ 1-bit measurements, with different values of $C_s$, $C'_s$ and NFEs. We mark **bold** the best scores, and underline the second best scores.

| | $C_s$ ($C'_s = 55$) | | | | | $C'_s$ ($C_s = 55$) | | | | | NFE ($C_s = 55$, $C'_s = 55$) | | | |
|---|---|---|---|---|---|---|---|---|---|---|---|---|---|---|
| Value | 50 | 52 | 55 | 58 | **60** | **50** | 52 | 55 | 58 | 60 | 20 | 50 | 80 | 100 |
| PSNR (↑) | 14.441 | 14.625 | 14.860 | 14.956 | 14.971 | **15.337** | 15.214 | 14.860 | 14.451 | 14.205 | 14.860 | 14.860 | 14.860 | 14.860 |
| | ±1.037 | ±1.119 | ±1.201 | ±1.275 | **±1.334** | **±1.354** | ±1.252 | ±1.201 | ±1.185 | ±1.146 | ±1.201 | ±1.201 | ±1.201 | ±1.201 |
| SSIM (↑) | 0.427 | 0.442 | 0.460 | 0.467 | 0.470 | **0.481** | 0.475 | 0.460 | 0.440 | 0.426 | 0.460 | 0.460 | 0.460 | 0.460 |
| | ±0.096 | ±0.099 | ±0.103 | ±0.106 | **±0.108** | **±0.107** | ±0.104 | ±0.103 | ±0.101 | ±0.099 | ±0.103 | ±0.103 | ±0.103 | ±0.103 |
| LPIPS (↓) | 0.444 | 0.436 | 0.433 | 0.441 | 0.447 | 0.440 | 0.436 | **0.433** | 0.440 | 0.447 | 0.433 | 0.433 | 0.433 | 0.433 |
| | ±0.090 | ±0.090 | **±0.094** | ±0.097 | ±0.099 | ±0.100 | ±0.098 | **±0.094** | ±0.091 | ±0.090 | ±0.094 | ±0.094 | ±0.094 | ±0.094 |

## G. Ablation Study for SIM-DMIS

In this section, we present a detailed analysis of the ablation experiments conducted on `SIM-DMIS` using the CIFAR-10 dataset with a measurement of $m = 500$. The experimental setup follows the same framework as in the main body of the research, and the DDIM sampling and DM2M inversion technique is applied throughout. The results are shown in Table 8.

When varying $C_s$ while fixing $C'_s = 55$ and NFE = 50, the trend of PSNR and SSIM are consistent with the `SIM-DMS` where higher values of $C_s$ also led to better signal quality and structural similarity. On the other hand, when $C_s$ is fixed at 55 and $C'_s$ is varied, the trends for PSNR and SSIM are largely similar to the `SIM-DMS` results, where lower values of $C'_s$ lead to better performance in these metrics. Interestingly, while the improvement in PSNR and SSIM is clear, LPIPS remains relatively stable across a wide range of $C'_s$ values, with a slight decrease at lower values of $C'_s$, particularly around 50 and 52. Finally, when adjusting the NFE with $C_s = 55$ and $C'_s = 55$, the results show minimal fluctuation in all performance metrics. This observation is consistent with the findings from `SIM-DMS`, where a similar behavior was noted.

In summary, the findings from the ablation study of `SIM-DMIS` underline several important insights. The parameter $C_s$ has a notable impact on both the signal quality and structural similarity of the reconstructed images, with higher values leading to better PSNR and SSIM. The parameter $C'_s$ plays a crucial role in enhancing PSNR and SSIM, with lower values yielding better results, while LPIPS remains stable. Additionally, the minimal effect of increasing NFE suggests that `SIM-DMIS` is highly efficient and provides stable performance even with fewer evaluations.

*Table 8.* Ablation experiments for `SIM-DMIS` on CIFAR-10 with $m = 500$ 1-bit measurements. We mark **bold** the best scores, and underline the second best scores.

| | $C_s$ ($C_s' = 55$, NFE=150) | | | | | $C_s'$ ($C_s = 55$, NFE=150) | | | | | NFE ($C_s = 55$, $C_s' = 55$) | | | |
|---|---|---|---|---|---|---|---|---|---|---|---|---|---|---|
| Value | 50 | 52 | 55 | 58 | 60 | 50 | 52 | 55 | 58 | 60 | 100 | 150 | 200 | 250 |
| PSNR ($\uparrow$) | 15.388 | 15.562 | 15.733 | 15.809 | 15.820 | **16.247** | 16.091 | 15.733 | 15.322 | 15.057 | 15.733 | 15.733 | 15.733 | 15.733 |
| | $\pm1.218$ | $\pm1.331$ | $\pm1.420$ | $\pm1.513$ | $\pm1.592$ | $\pm1.655$ | $\pm1.500$ | $\pm1.420$ | $\pm1.355$ | $\pm1.307$ | $\pm1.420$ | $\pm1.420$ | $\pm1.420$ | $\pm1.420$ |
| SSIM ($\uparrow$) | 0.489 | 0.506 | 0.522 | 0.531 | 0.534 | **0.548** | 0.540 | 0.522 | 0.500 | 0.484 | 0.522 | 0.522 | 0.522 | 0.522 |
| | $\pm0.097$ | $\pm0.098$ | $\pm0.099$ | $\pm0.101$ | $\pm0.104$ | $\pm0.101$ | $\pm0.099$ | $\pm0.099$ | $\pm0.099$ | $\pm0.099$ | $\pm0.099$ | $\pm0.099$ | $\pm0.099$ | $\pm0.099$ |
| LPIPS ($\downarrow$) | 0.406 | **0.399** | 0.402 | 0.411 | 0.417 | 0.409 | 0.405 | 0.402 | 0.404 | 0.411 | 0.402 | 0.402 | 0.402 | 0.402 |
| | $\pm0.095$ | $\pm0.093$ | $\pm0.095$ | $\pm0.100$ | $\pm0.102$ | $\pm0.103$ | $\pm0.098$ | $\pm0.095$ | $\pm0.093$ | $\pm0.094$ | $\pm0.095$ | $\pm0.095$ | $\pm0.095$ | $\pm0.095$ |

## H. Comparative Evaluation of SIM-DMS and SIM-DMIS on Performance and Efficiency

In this section, we evaluate the performance and efficiency of `SIM-DMS` and `SIM-DMIS` on the FFHQ dataset. The experimental setup follows the same framework as in the main body of the research. The results presented in the main paper (Table 1) show that `SIM-DMIS` consistently outperforms `QCS-SGM` and `SIM-DMFIS` in both reconstruction quality and NFE. Notably, `QCS-SGM` requires over 11,000 NFEs to achieve competitive results, incurring a high computational cost. Due to the performance of `SIM-DMIS` over both methods, and the particularly high computational cost of `QCS-SGM`, we exclude `QCS-SGM` and `SIM-DMFIS` from the comparisons in this section.

**Performance Evaluation** As discussed in Footnote 10 and Appendices F and G, ablation studies on CIFAR-10 show that further increasing the NFE for SIM-DMS does not lead to substantial performance enhancements.

To further verify this behavior and to compare the relative performance of `SIM-DMS` and `SIM-DMIS`, we conduct additional experiments on the FFHQ dataset. The results, summarized in Table 9, indicate that under the same NFEs, `SIM-DMIS` consistently outperforms `SIM-DMS` across all standard metrics.

*Table 9.* Quantitative results on FFHQ ($m = n/8 = 24576$). We mark **bold** the best scores, and underline the second best scores.

| Method | NFE | PSNR ($\uparrow$) | SSIM ($\uparrow$) | LPIPS ($\downarrow$) | FID ($\downarrow$) |
|---|---|---|---|---|---|
| `DPS-N` | 1000 | $11.14\pm1.46$ | $0.37\pm0.09$ | $0.69\pm0.05$ | 349.24 |
| `DPS-L` | 1000 | $8.57\pm2.05$ | $0.22\pm0.08$ | $0.69\pm0.09$ | 109.01 |
| `DAPS-N` | 1000 | $16.59\pm0.54$ | $0.33\pm0.05$ | $0.48\pm0.05$ | 138.70 |
| `DAPS-L` | 1000 | $5.63\pm0.71$ | $0.04\pm0.03$ | $0.61\pm0.03$ | 322.75 |
| `SIM-DMS` | 50 | $17.14\pm2.41$ | $0.44\pm0.07$ | $0.48\pm0.05$ | 105.04 |
| `SIM-DMS` | 150 | $17.72\pm2.63$ | $0.46\pm0.08$ | $0.48\pm0.06$ | 95.52 |
| `SIM-DMIS` | 50 | $18.78\pm3.09$ | $0.58\pm0.10$ | $0.41\pm0.08$ | 89.47 |
| `SIM-DMIS` | 150 | **$19.87\pm2.77$** | **$0.60\pm0.09$** | **$0.37\pm0.05$** | **76.21** |

**Reconstruction Speed** We conduct additional experiments to measure the reconstruction speed of `SIM-DMS` and `SIM-DMIS`. The inference time is averaged over 10 validation images from the FFHQ dataset. All experiments are executed on a single NVIDIA GeForce RTX 4090 GPU. As shown in Table 10, the results indicate that `SIM-DMS` and `SIM-DMIS` exhibit significantly faster reconstructions when compared to the competing methods.

*Table 10.* Comparison of inference time (in seconds) for reconstructing 10 images on FFHQ. We mark **bold** the best scores, and underline the second best scores.

| Method | NFE | Inference Time (s) ($\downarrow$) |
|---|---|---|
| DPS-N | 1000 | 142 |
| DPS-L | 1000 | 142 |
| DAPS-N | 1000 | 160 |
| DAPS-L | 1000 | 160 |
| SIM-DMS | 50 | **1.96** |
| SIM-DMIS | 150 | 5.66 |

