# OpenReview forum: "Learning Single Index Models with Diffusion Priors"
_ICML.cc/2025/Conference — ICML 2025 poster_

### Official Review · Reviewer_3BCR · 2025-02-18

**Overall Recommendation:** 3

**Summary:**

The work addresses the problem of signal reconstruction in semi-parametric single-index models, where the link function is unknown. They propose a new method relying on parametrizing the signal prior by a diffusion model. Building on the observation that the measurements may be related to noisy versions of the signal, the related noise variance (and corresponding diffusion time) is learned in the method, and the generator of the diffusion process and its inverse used to yield the final estimator. Theoretical learning guarantees are derived. Numerical experiments on the FFHQ and ImageNet datasets are provided, revealing competitivity of the method compared to existing schemes.

**Claims And Evidence:**

The method is benchmarked against concurrent schemes, in terms of performance metrics and compute. Reconstructed images by the evaluated methods are further illustrated in Figs. 2,3.

**Essential References Not Discussed:**

I did not identify any essential missing reference.

**Experimental Designs Or Analyses:**

I did not check the details of the experimental designs.

**Methods And Evaluation Criteria:**

To the best of my awareness, the benchmark datasets considered were also selected in previous works on the topic to evaluate the methods. For instance, FFHQ was used in (Meng and Kabashima, 2023), to benchmark the QCS-SGM method. Therefore, they seem relevant for the evaluation, and ensure comparability with some of the previous works.

**Other Comments Or Suggestions:**

I do not have further comments or suggestions.

**Other Strengths And Weaknesses:**

The paper is well written, and intuition is provided in support of the method. I have a few clarifying questions, which I included in the Theoretical Claims section. I am overall in favor of acceptance.

**Questions For Authors:**

Some clarifying questions are detailed in previous sections.

**Relation To Broader Scientific Literature:**

Although I have limited familiarity with the literature, it is my understanding that compared to key previous works such as (Meng and Kabashima, 2023) or (Zhang et al., 2024), the main novelty of the method is its ability to accommodate unknown non-linear link functions. It also displays competitive performance with previously proposed methods. However, it is possible there exists other methods which I may be missing.

As a minor comment, the related work section could be further clarified, notably the last paragraph of page 2. Currently, it is unclear which work address linear or non-linear settings, and in which works the link function is (un)known.

**Theoretical Claims:**

I did not check carefully the proofs of the theoretical claims.

To strengthen the paper, it would be beneficial to include a version of Theorem 2 for the SIM-DMS algorithm (28), if feasible. For now, only the SIM-DMIS algorithm is endowed with the theoretical guarantees. Besides, Theorem 2 does not bear entirely on the SIM-DMIS estimator. Would it be feasible to combine (24) and (29) to reach an end-to-end guarantee?

---

> ### Author Rebuttal · Authors · 2025-03-31
>
> Thanks for your recognition of this paper and the helpful comments and suggestions. Our responses to the main concerns are as follows. All citations refer to the reference list in the main document.
>
> (**To strengthen the paper, it would be beneficial to include a version of Theorem 2 for the SIM-DMS algorithm (28), if feasible. For now, only the SIM-DMIS algorithm is endowed with the theoretical guarantees. Besides, Theorem 2 does not bear entirely on the SIM-DMIS estimator. Would it be feasible to combine (24) and (29) to reach an end-to-end guarantee?**)
> We are grateful to the reviewer for the valuable suggestions. In the revised paper, we will include a version of Theorem 2 for SIM-DMS. Eq. (29) within the statement of Theorem 2 essentially says that for any sample drawn from the marginal distribution $q_t$, when we perform the inversion process from time $t$ to $T$, followed by the sampling process from time $T$ to $\epsilon$, the resulting vector will be reconstructed to closely approximate the corresponding ground-truth data from $q_0$ (and lie on the same ODE trajectory as the original sample). Additionally, Eq. (24) (along with Eq. (25)) essentially indicates that  (the scaled version of) $\frac{\mathbf{A}^T\mathbf{y}}{m}$ approximately follows the marginal distribution $q_{t^*}$ for some $t^*$. If we make the (relatively strong) assumption that (the scaled version of) $\frac{\mathbf{A}^T\mathbf{y}}{m}$ precisely follows $q_{t^*}$ for some $t^*$, then combining Eqs. (24) and (29) provides an end-to-end guarantee. We leave the end-to-end guarantee in the case where Eq. (25) only approximately holds to future study.
>
> (**As a minor comment, the related work section could be further clarified, notably the last paragraph of page 2. Currently, it is unclear which work address linear or non-linear settings, and in which works the link function is (un)known.**)
> Among the methods described in the last paragraph of page 2, MCG (Chung et al., 2022b) and $\Pi$GDM (Song et al., 2023) are mainly designed for the linear setting. Additionally, $\Pi$GDM can be extended to certain nonlinear settings (where the link function is known) by leveraging a combination of pseudoinverse operations and nonlinear transformations. DPS (Chung et al., 2023b) and DAPS (Zhang et al., 2024) are applicable in nonlinear settings as well, also primarily under the assumption that the link function is known. We will clarify these in the revised version.

---

> > ### Comment · Reviewer_3BCR · 2025-04-02
> >
> > I thank the authors for clarifying these points and promising the revisions. The answers seem to corroborate my current evaluation and understanding of the paper, and I maintain my score.

---

### Official Review · Reviewer_g5pJ · 2025-03-13

**Overall Recommendation:** 3

**Summary:**

The authors propose a diffusion model sampling scheme to reconstruct an unknown signal from measurements, assuming a single-index model with a known compressed sensing matrix and noise distribution but unknown and potentially nondifferentiable link function. The approach leverages a property of the link function to find an intermediate time at which to begin inversion, thereby making inversion much more efficient than the naïve approach. The authors provide a theoretical analysis and numerical experiments on image datasets, comparing to state-of-the-art diffusion inverse solvers.

**Claims And Evidence:**

Yes.

**Essential References Not Discussed:**

No.

**Experimental Designs Or Analyses:**

Yes, for the 1-bit compressed sensing experiments on FFHQ and ImageNet. It is strange that DPS and DAPS with knowledge of the link function (i.e., DPS-N and DAPS-N) perform worse than they do without knowledge of the link function (i.e., DPS-L and DPS-L). This is true for all the results in Tables 1 and 2 except DPS in Table 2.

**Methods And Evaluation Criteria:**

Yes.

**Other Comments Or Suggestions:**

N/A

**Other Strengths And Weaknesses:**

[Strengths] The proposed idea is quite simple and makes for an efficient algorithm. It leverages a property of the link function (Eq. 20), so the simple algorithm has some theoretical inspiration behind it.

[Weaknesses] The method makes several shortcuts without strong theoretical justifications for them. For example, the leap from Eq. 27 to Eq. 28 is rather tenuous if it relies on the assumption that the reverse diffusion process will effectively remove the noise from the naïve estimate. Overall, it is difficult to tell how theoretically justified the method is. There is little intuitive explanation of Theorem 2 (I could not find a Theorem 2, by the way), making it difficult to judge its significance and usefulness. And there is no discussion about whether this would provide samples from a Bayesian posterior.

The presentation of the paper is rather poor. The authors should include more intuitive explanations. For example, what exactly are the properties in Eqs. 20 and 21 saying? What is the intuition behind Eq. 25 and why it’s different from Eq. 22? Also, the background on diffusion models is too dense and contains a lot of unnecessary information. I recommend sticking the equations and ideas that are essential just for the understanding of this paper.

It is strange that DPS-N and DAPS-N perform so poorly in the numerical experiments. I am curious why the authors think they would perform so poorly compared to SIM-DMS and SIM-DMIS.

**Questions For Authors:**

N/A

**Relation To Broader Scientific Literature:**

One perspective is that this work extends the work of Meng and Kabashima 2022 to handle unknown link functions. Another perspective is that it adds to the vast literature of diffusion inverse solvers (including DPS, DAPS, and Pi-GDM) with a simple method for a particular type of inverse problem, namely one where the measurements are an unknown/nondifferentiable transformation of compressed sensing measurements.

**Theoretical Claims:**

I was unable to check proof correctness.

---

> ### Author Rebuttal · Authors · 2025-03-31
>
> Thanks for your helpful comments and suggestions. Our responses to the main concerns are as follows. All citations refer to the reference list in the main document.
>
> (**It is strange that DPS and DAPS with knowledge of the link function (i.e., DPS-N and DAPS-N) perform worse than they do without knowledge of the link function (i.e., DPS-L and DAPS-L).**)
> In 1-bit measurements, even in the noiseless scenario, the link function $f(x) = \mathrm{sign}(x)$ is non-differentiable at $x = 0$ (though as mentioned in Footnote 9, PyTorch can still enforce automatic differentiation). The differentiability of $f$ matters significantly. As mentioned in Section 1.1, under the strong differentiability assumption on the link function, (Wei et al., 2019) can handle general non-Gaussian sensing vectors. However, most SIMs research assumes Gaussian sensing vectors, like the seminal work (Plan & Vershynin, 2016) and many follow-up studies such as (Liu & Liu, 2022). Without differentiability, extending these to non-Gaussian cases is difficult.
>
> The non-differentiability also poses challenges for DPS-N and DAPS-N as they rely on $f$ in gradient based updates. This can lead to inaccurate gradients and ultimately resulting in subpar performance. (The DAPS supplementary material suggests using Metropolis Hasting for non-differentiable forward operators, but it is also mentioned to have inferior performance and low efficiency, with results only in the supplementary material.) In contrast, DPS-L and DAPS-L do not utilize $f$, thus allowing for relatively better performance (note that as demonstrated in the work (Plan & Vershynin, 2016), SIMs can be transformed into linear measurement models with unconventional noises).
>
> (**There is little intuitive explanation of Theorem 2 (I could not find a Theorem 2, by the way). And there is no discussion about whether this would provide samples from a Bayesian posterior.**)
> Theorem 2 essentially says that for any sample drawn from the marginal distribution $q_t$, when we perform the inversion process from time $t$ to $T$, followed by the sampling process from time $T$ to $\epsilon$, the resulting vector will be reconstructed to closely approximate the corresponding ground-truth data from $q_0$ (and lie on the same ODE trajectory as the original sample). In the revised version, we will incorporate this intuitive explanation in the paragraph preceding Theorem 2.
> We are unsure what the reviewer means by “I could not find a Theorem 2”. We guess that the reviewer might be referring to the proof of Theorem 2, which is available in Appendix B.1.
>
> The approaches presented in our work are not directly related to sampling from a Bayesian posterior. While it would be interesting to discuss whether our approaches could yield samples from a Bayesian posterior, we believe that such an exploration is beyond the scope of the current work.
>
> (**The authors should include more intuitive explanations. For example, what exactly are the properties in Eqs. 20 and 21 saying? What is the intuition behind Eq. 25 and why it’s different from Eq. 22? Also, the background on diffusion models is too dense and contains a lot of unnecessary information.**)
> The condition in Eq. (20) is a classic and crucial condition for SIMs. For example, it is (albeit implicitly) assumed in the seminal work (Plan & Vershynin, 2016) and in subsequent research that builds upon it. If this condition fails to hold, specifically when $\mu  = 0$, the recovery of $\mu \mathbf{x}^*$ as in (Plan & Vershynin, 2016) and in our Eq. (24) becomes meaningless. We follow (Liu & Liu, 2022) to assume the condition in Eq. (21), which generalizes the assumption that $f(\mathbf{a}^T\mathbf{x}^*)$ is sub-Gaussian (which is satisfied by quantized measurement models), and accommodates more general nonlinear measurement models, such as cubic measurements with $f(x)=x^3$ and their noisy counterparts.
>
> The intuition underlying Eq. (25) is that (the scaled version of) $\frac{\mathbf{A}^T\mathbf{y}}{m}$ is approximately the ground-truth signal $\mathbf{x}^*$ (drawn from the target data distribution $q_0$) with an added zero-mean Gaussian noise component. Then, considering the forward process of diffusion models, which gradually adds Gaussian noise to the ground-truth data, performing a full inversion (from time $\epsilon$ to $T$) and then a full sampling from time $T$ to $\epsilon$ as in Equation (22) is inappropriate. As mentioned in the paragraph below Eq. (22), this approach implicitly (and wrongly) assumes that $\frac{\mathbf{A}^T\mathbf{y}}{m}$ approximately follows the target data distribution $q_0$ (without taking additive zero-mean Gaussian noise into account).
>
> In the revised version, we will incorporate the above intuitive explanations. Additionally, we will streamline the background on diffusion models, focusing solely on the equations and concepts that are fundamental to understanding the core ideas presented in this paper.

---

### Official Review · Reviewer_68Nk · 2025-03-13

**Overall Recommendation:** 4

**Summary:**

In this manuscript, the authors address a notable shortcoming in current signal recovery techniques based on diffusion models: most existing methods either concentrate on narrowly defined reconstruction tasks or fail to handle nonlinear measurement models with discontinuous or unknown link functions. To tackle this issue, the authors propose an efficient reconstruction strategy that requires only a single round of unconditional sampling and partial inversion of the diffusion models. Their theoretical analysis and experimental evaluations collectively verify the effectiveness of this approach.

**Claims And Evidence:**

Yes.

**Essential References Not Discussed:**

No.

**Experimental Designs Or Analyses:**

I have carefully reviewed the experimental results and noticed that the SIM-DMS method achieves the second-best performance with only 50 NFE, whereas SIM-DMIS requires 150 NFE. To ensure a fair and comprehensive comparison, I suggest conducting additional experiments, specifically comparing SIM-DMIS with 50 NFE and SIM-DMS with 150 NFE. This would help better understand the relative performance under equivalent NFE.

**Methods And Evaluation Criteria:**

FID is widely recognized as a key evaluation metric in diffusion models. Why wasn’t it employed here?

**Other Comments Or Suggestions:**

1. It would be beneficial to include additional comparisons highlighting the reconstruction speed of the proposed method relative to existing approaches.
2. Some terms in the references need to maintain consistency with the original papers, particularly regarding capitalization. For instance, on line 500, "ImageNet" should be capitalized exactly as in the original source. Please carefully check and correct all similar cases to ensure accurate citation formatting.

**Other Strengths And Weaknesses:**

Strengths:

1. The paper is clearly structured and well-organized.
2. The experimental evaluations presented are comprehensive.
3. The theoretical derivations provided are rigorous and complete.

Weaknesses:

1. While the SIM-DMS method achieves competitive performance using only 50 neural function evaluations (NFEs), the SIM-DMIS approach requires 150 NFEs. To ensure a fair comparison, the authors are encouraged to perform additional experiments by evaluating SIM-DMIS with 50 NFEs and SIM-DMS with 150 NFEs. This would provide a clearer understanding of the relative performance of both methods under an equivalent NFEs.

**Questions For Authors:**

FID is widely recognized as a key evaluation metric in diffusion models. Why wasn’t it employed here?

**Relation To Broader Scientific Literature:**

This work extends the approach of QCS-SGM by addressing its limitations. While QCS-SGM either focuses on specific reconstruction problems or cannot effectively handle nonlinear measurement models with discontinuous or unknown link functions, the proposed method overcomes these challenges. Moreover, the proposed method achieves more accurate reconstructions with significantly fewer neural function evaluations (NFEs) compared to QCS-SGM.

**Theoretical Claims:**

I have reviewed the "Setup and Approaches" section and did not identify any issues.

---

> ### Author Rebuttal · Authors · 2025-03-31
>
> Thanks for your recognition of this paper and the helpful comments and suggestions. Our responses to the main concerns are as follows.
>
> (**FID is widely recognized as a key evaluation metric in diffusion models. Why wasn’t it employed here?**)
> We carry out additional experiments to report the FID. The results are presented in Table B1 below. Given the time constraint during the rebuttal period, and following the work for DAPS (Zhang et al., 2024), we compute the FID using a set of 100 validation images. The FID results for QCS-SGM, DPS, and DAPS are not included at this stage as their calculation is relatively time-consuming. However, in the revised version, we will incorporate the FID results for all methods.
>
> (**To ensure a fair and comprehensive comparison, I suggest conducting additional experiments, specifically comparing SIM-DMIS with 50 NFE and SIM-DMS with 150 NFE.**)
> Thanks for the insightful comment. We have conducted simple ablation studies for SIM-DMS and SIM-DMIS on the CIFAR-10 dataset (mentioned in Footnote 10, and detailed in Appendices F and G), and the results show that further increasing the NFE for SIM-DMS does not lead to substantial performance enhancements. We also perform the suggested experiments on the FFHQ dataset, and summarize the additional results in the following table. The results also indicate that under the same NFEs, SIM-DMIS consistently outperforms SIM-DMS across all metrics.
>
> | Method   | NFE | PSNR          | SSIM          | LPIPS         | FID    |
> |----------|-----|---------------|---------------|---------------|--------|
> | SIM-DMS  | 50  | 17.14 ± 2.41  | 0.44 ± 0.07   | 0.48 ± 0.05   | 105.04 |
> | SIM-DMS  | 150 | 17.72 ± 2.63  | 0.46 ± 0.08   | 0.48 ± 0.06   | 95.52  |
> | SIM-DMIS | 50  | 18.78 ± 3.09  | 0.58 ± 0.10   | 0.41 ± 0.08   | 89.47  |
> | SIM-DMIS | 150 | 19.87 ± 2.77  | 0.60 ± 0.09   | 0.37 ± 0.05   | 76.21  |
>
> Table B1: Quantitative comparison of SIM-DMS and SIM-DMIS on the FFHQ dataset.
>
> (**It would be beneficial to include additional comparisons highlighting the reconstruction speed of the proposed method relative to existing approaches.**)
> We conduct additional experiments to measure the reconstruction speed of our method, and the results are presented in the table below. The reported inference time refers to the average reconstruction time for 10 validation images from the FFHQ dataset. All of these experiments are executed on a single NVIDIA GeForce RTX 4090 GPU. The results indicate that SIM-DMS and SIM-DMIS exhibit significantly faster reconstructions when compared to the competing methods (we do not compare with QCS-SGM since it is very time-consuming). We will include the comparisons with respect to reconstruction speed in the revised version.
>
> | Method   | NFE  | Inference Time (s) |
> |----------|------|--------------------|
> | DPS-N    | 1000 | 142                |
> | DPS-L    | 1000 | 142                |
> | DAPS-N   | 1000 | 160                |
> | DAPS-L   | 1000 | 160                |
> | SIM-DMS  | 50   | 1.96               |
> | SIM-DMIS | 150  | 5.66               |
>
> Table B2: Comparisons for reconstruction speed on the FFHQ dataset.
>
> (**Some terms in the references need to maintain consistency with the original papers, particularly regarding capitalization.**)
> In our revised version, we will carefully check all the references and correct all the inconsistent cases to ensure accurate citation formatting.

---

### Official Review · Reviewer_H7Q4 · 2025-03-24

**Overall Recommendation:** 1

**Summary:**

Summary
This paper proposes a novel method for reconstructing images from measurements obtained through a nonlinear compressed sensing model. The degradation model consists of a measurement matrix, an unknown and potentially discontinuous nonlinear element-wise link function, and additive Gaussian noise.

The proposed reconstruction algorithm begins by initializing the estimate using the measurement vector, applying the transposed measurement matrix, and applying an empirically tuned normalization. The final reconstruction is achieved using a pre-trained diffusion model (DM). The method involves partial DM inversion followed by DM sampling, where the inversion start time is determined based on the norm of the DM inversion initialization. The authors provide a theoretical analysis by proving a theorem that offers an upper bound on the distance between the reconstructed image and the desired image under specific conditions.

Main findings:
- The authors establish an upper bound for the distance between the reconstructed and the desired images.
- Experimental results demonstrate that the proposed approach outperforms existing methods in 1-bit and cubic measurement scenarios.
- The authors show that the combination of partial inversion and sampling yields better reconstruction quality compared to sampling alone or full inversion followed by sampling.

## Update After Rebuttal

I thank the authors for their detailed responses. I respectfully disagree with the author's assumption that the constant $C$ is fixed and independent of the number of inversion steps (i.e., the number of times the denoiser is applied). This assumption leads to an unrealistic conclusion that increasing the number of inversion steps indefinitely would drive the upper bound on the reconstruction error to zero, thereby implying perfect reconstruction. If this were the case, the authors should demonstrate in the experimental section that the reconstruction error consistently decreases toward zero as the number of inversion steps increases.

A more reasonable and realistic assumption is that the denoiser has a fixed Lipschitz constant **per activation**, in which case the overall constant $C$ in the bound would grow with the number of inversion steps.

Furthermore, in contexts involving bounded quantities, such as image pixels constrained within the range $[0, 1]$, expressing an upper bound using capital-O notation such as $O(1)$ is inappropriate. Instead, the upper bound should be stated explicitly as $distance \le C$, where $C$ is a meaningful, domain-aware constant (in the case of image pixels less than 1). As noted in my original review, proving that the **per-pixel** distance between the reconstructed and desired images is $O(1)$, which allows for an arbitrary constant, potentially even exceeding 1, does not yield a meaningful or non-trivial insight in this context.

Given the concerns outlined above, I maintain my original recommendation to reject the manuscript.

**Claims And Evidence:**

I the claims made in the submission are supported by evidence.

**Essential References Not Discussed:**

I did not come across related works that are essential for understanding the key contributions of the paper but are not currently cited.

**Experimental Designs Or Analyses:**

In my opinion, the experiments presented in the manuscript are sound

**Methods And Evaluation Criteria:**

In my opinion, yes.

**Other Comments Or Suggestions:**

I have no further comments.

**Other Strengths And Weaknesses:**

Since the theoretical analysis is a significant component of the manuscript's contribution, I recommend rejecting the manuscript.

**Questions For Authors:**

I have no further questions for the authors.

**Relation To Broader Scientific Literature:**

The key contributions of the manuscript are related to the literature about image reconstruction using diffusion models.

**Theoretical Claims:**

I have a concern regarding the theoretical part of the manuscript. The upper bound presented in the main theorem appears trivial and lacks practical significance. The theorem states that:

$\|\|x - x_r\|\|_2 = O(\sqrt{n}(h^1 + Lh^2))$,

where $x$ is the desired image, $x_r$ is the reconstructed image, $n$ is the image dimension, $L$ is a Lipschitz constant, and $h^1$ and $h^2$ are diffusion model parameters.

This result essentially implies:

$\|\|x - x_r\|\|_2 \le Ch\sqrt{n}$,

where $C$ is a positive constant and $h =  h^1 + Lh^2$.

However, the image pixels are typically bounded within the range [0, 1]. Therefore, the maximum possible Euclidean distance between any two images is already bounded by $\sqrt{n}$, representing the distance between a completely black and a completely white image. Consequently, the upper bound derived in the theorem does not provide any meaningful or non-trivial insight.

If the pixel were unbounded, the theorem might offer valuable insights. However, under the current setting, the theoretical contribution seems to lack practical relevance.

---

> ### Author Rebuttal · Authors · 2025-03-31
>
> We thank the reviewer for the feedback. We are pleased that the claims in our submission have been recognized as "supported by evidence" and our experiments have been recognized as "sound". Regarding your major concern about the practical relevance of Theorem 2, our responses are as follows:
> -  For practical relevance, the parameter $h_{\max}=\max_{i \in [N]} \big(\lambda_{t_i}-\lambda_{t_{i-1}}\big)$ in Eq. (29) plays an important role. When the number of sampling/inversion steps is large, for instance, on the order of $10^2$ or $10^3$, $h_{\max}$ is approximately $10^{-2}$ or $10^{-3}$ (if using typical uniform $\lambda$ steps). Therefore, when the number of sampling/inversion steps is sufficiently large, in the upper bound $C h \sqrt{n}$ illustrated by the reviewer, the value of $h$ will be much smaller than $1$ (e.g., $h = 0.001$; note that $C$ is a positive constant of order $\Theta(1)$). This makes the upper bound practically meaningful when image pixels are bounded within the range $[0,1]$.
> - The proof of Theorem 2 (see Appendix B.1) is built upon the proof for Theorem 3.2 in the popular work for DPM-Solver (Lu et al., 2022a) (note that the upper bound in (Lu et al., 2022a, Theorem 3.2) is for each individual pixel, and thus their bound does not have the $\sqrt{n}$ factor), which is in turn based on classic analyses of local truncation error for numerical ODE solvers, such as those presented in Section 5.6 of (Burden & Faires, 2005). And our Theorem 2 bears similar practical relevance to the theoretical findings presented in these works.
>
>
> Richard L. Burden and J. Douglas Faires, Numerical Analysis, 8th edition, Thomson/Brooks/Cole, 2005.

---

> > ### Comment · Reviewer_H7Q4 · 2025-04-04
> >
> > According to the definition provided in the manuscript [*], Section 1.3 Notation, lines 114-117, is it correct to conclude that any arbitrary constant $C$, no matter how large, for example, $C = 1000$, is considered to be of order $O(1)$?
> >
> > [*] "Given two sequences of real values $\{a_i\}$ and $\{b_i\}$, we write $a_i = O(b_i)$ if there exists an absolute constant $C_1$ and a positive integer $i_1$ such that for any $i > i_1$, $|a_i| ≤ C_1b_i$."

---

> > > ### Author Response · Authors · 2025-04-06
> > >
> > > Thank you for the update. Our responses are presented as follows. All citations refer to the reference list at the end of these responses.
> > >
> > > The implied constant $C$ within the $O(\cdot)$ term in our Eq. (29) is similar to the one in the $O(\cdot)$ term stated in (Lu et al., 2022, Theorem 3.2). This $C$ is a fixed positive constant that depends on the pre-trained neural network model; it is not a variable parameter that can grow arbitrarily large. In contrast, $h$ is a variable parameter that can approach zero. The upper bound $C h \sqrt{n}$ can be interpreted as follows: Given $\varepsilon \in (0,1)$, to achieve pixel-wise $\varepsilon$-accuracy, $h$ only needs to be smaller than $\frac{\varepsilon}{C}$. For instance, if $C$ is fixed at $1000$ (and remains constant thereafter) and $\varepsilon=0.1$, $h$ only needs to be smaller than $0.0001$. Although this may seem to demand that $h$ be very small (or the number of sampling/inversion steps be very large), we note that upper bounds similar to ours are prevalent in the theoretical analysis of diffusion models. Examples include the bound in (Lu et al., 2022) and those in recent theoretical works such as (Chen et al., 2023a; Chen et al., 2023b; Chen et al., 2023c; Li et al., 2024).
> > >
> > > For a specific example, to achieve $\varepsilon$-accuracy in terms of the total variation distance, (Li et al., 2024, Theorem 1) requires the number of steps to exceed $C\big(\frac{n^2}{\varepsilon} + \frac{n^3}{\sqrt{\varepsilon}}\big)$, where $C$ is a fixed positive constant and some minor logarithmic terms have been omitted. In practical applications, the data dimension $n$ is often very large. For instance, for the FFHQ 256x256 dataset, $n = 256\times 256 \times 3 = 196608$. This example directly follows the bound in (Li et al., 2024), and similarly implies a very large number of sampling steps in high-dimensional settings, indicating that such bounds have been widely accepted in the research area of theoretical analyses of diffusion models.
> > >
> > > References:
> > >
> > > [1] Lu et al. "DPM-Solver: A fast ODE solver for diffusion probabilistic model sampling in around 10 steps." NeurIPS, 2022. [1354 citations]
> > >
> > > [2] Chen et al. “Sampling is as easy as learning the score: theory for diffusion models with minimal data assumptions." ICLR, 2023. [325 citations]
> > >
> > > [3] Chen et al. “Improved analysis of score-based generative modeling: User-friendly bounds under minimal smoothness assumptions." ICML, 2023. [177 citations]
> > >
> > > [4] Chen et al. “The probability flow ODE is provably fast." NeurIPS, 2023. [108 citations]
> > >
> > > [5] Li et al. "Towards non-asymptotic convergence for diffusion-based generative models." ICLR, 2024. [92 citations in total for two versions of the work as shown in the first author’s Google Scholar page]

---

### Decision · Program_Chairs · 2025-05-01

**Decision:**

Accept (poster)

**Comment:**

The paper proposes a compressed sensing inversion algorithm based on diffusion modeling. Three reviewers thought the paper was good and should be accepted, while one argued for rejection. The basis for rejection of this reviewer was a perceived triviality of a theorem in the paper, but the authors argue convincingly that this is a misunderstanding on the part of the reviewer and is in line with similar results in compressed sensing theory. Considering this, I am inclined to consider the three positive reviews as reflecting the contributions of this paper, and these reviewers appreciated the comparison with state of art diffusion solvers that indicated the proposed inversion algorithm performs well compared to these.

One suggestion to consider for the paper was to make it a bit more reader friendly by introducing intuitive examples, etc. While I don't consider this to be required for acceptance, it is useful to note that the paper doesn't truly "begin" until well into the third page with an overly extensive review of related work, which in general is unnecessary. As a result, the experiments are crammed into the last page. Considering the extensive experiments (plus more theory) shown in the appendix, I think the authors have done a more comprehensive evaluation than shown in the main paper and they may consider incorporating some of this into the experiments section and reducing the introductory material in order to balance the main paper.

Overall, I'm inclined to recommend acceptance of the paper considering the positive reviews and more extensive experiments included in the appendix. The theoretical results are also in line with what would be appreciated by the ICML audience.